# Astrocyte aquaporin mediates a tonic water efflux maintaining brain homeostasis

Cuong Pham[1], Yuji Komaki[2], Anna Deàs-Just[1], Benjamin Le Gac[1], Christine Mouffle[1], Clara Franco[1], Agnès Chaperon[1], Vincent Vialou[1], Tomokazu Tsurugizawa[3,4]*, Bruno Cauli[1]*, Dongdong Li[1]*

[1]Sorbonne Université - CNRS - INSERM, Institut de Biologie Paris Seine, Neuroscience Paris Seine, Paris, France; [2]Central Institute for Experimental Medicine and Life Science, Kawasaki, Japan; [3]Human Informatics and Interaction Research Institute, National Institute of Advanced Industrial Science and Technology (AIST), Tsukuba, Japan; [4]Faculty of Engineering, University of Tsukuba, Tsukuba, Japan

**\*For correspondence:**
t-tsurugizawa@aist.go.jp (TT);
bruno.cauli@upmc.fr (BC);
dongdong.li@inserm.fr (DL)

**Competing interest:** The authors declare that no competing interests exist.

## eLife Assessment

In this work, the authors propose that astrocytic aquaporin 4 (AQP4) is the main pathway for tonic water efflux, without which astrocytes undergo cell swelling. These findings are **important**, because they shed light on key molecular mechanisms implicated with the regulation of brain water homeostasis. The authors use a broad set of experimental tools (e.g., acute brain slices, in vivo recording, and diffusion-weighted MRI) but the evidence remains **incomplete** without ruling out non-specific effects of TGN-020, and without evidence that changes in sulforhodamine B fluorescence can be used as reliable readouts of cell volume dynamics.

**Abstract** Brain water homeostasis not only provides a physical protection, but also determines the diffusion of chemical molecules key for information processing and metabolic stability. As a major type of glia in brain parenchyma, astrocytes are the dominant cell type expressing aquaporin water channel. How astrocyte aquaporin contributes to brain water homeostasis in basal physiology remains to be understood. We report that astrocyte aquaporin 4 (AQP4) mediates a tonic water efflux in basal conditions. Acute inhibition of astrocyte AQP4 leads to intracellular water accumulation as optically resolved by fluorescence-translated imaging in acute brain slices, and in vivo by fiber photometry in mobile mice. We then show that aquaporin-mediated constant water efflux maintains astrocyte volume and osmotic equilibrium, astrocyte and neuron $Ca^{2+}$ signaling, and extracellular space remodeling during optogenetically induced cortical spreading depression. Using diffusion-weighted magnetic resonance imaging (DW-MRI), we observed that in vivo inhibition of AQP4 water efflux heterogeneously disturbs brain water homeostasis in a region-dependent manner. Our data suggest that astrocyte aquaporin, though bidirectional in nature, mediates a tonic water outflow to sustain cellular and environmental equilibrium in brain parenchyma.

## Introduction

Every aspect of brain function relies on the delicately maintained water environment. It supports brain structural stability and molecular diffusion, laying the ground for information processing, metabolite shuttling, and adaptation to living environments (*Kimelberg, 2004*). Water is the fundamental constituent of the cerebrospinal fluid (CSF) infiltrating into the central nervous system (CNS) and the

interstitial fluid distributed in brain parenchyma (*Agnati et al., 2017*; *Brinker et al., 2014*). Brain fluid transport is suggested to support the diffusion of energetic fuels like glucose and lactate to warrant the metabolic circumstances in the parenchyma, and is also implicated in the clearance of waste molecules from the brain as described in the glymphatic system (*Abbott et al., 2018*; *Iliff et al., 2013*; *Iliff et al., 2012*; *Mestre et al., 2018a*; *Rasmussen et al., 2022*). In addition, the restricted mobility of water molecules diffusion within intra- and extracellular space reflects the brain microstructure. The diffusion-weighted magnetic resonance imaging (DW-MRI) detects the mobility of water proton for structural and functional neuroimaging (*Le Bihan et al., 2006*; *Tsurugizawa et al., 2013*), being adopted for clinical diagnostics in neurological diseases such as ischemia, brain tumor, and edema (*Gaddamanugu et al., 2022*; *Le Bihan and Iima, 2015*) and recently leveraged for assessing brain glymphatic system (*Giannetto et al., 2024*; *Gomolka et al., 2023*).

Water equilibrium in the brain is maintained by dynamic transport between different cell entities. Aquaporin is a family of transmembrane channels facilitating bidirectional water flow, with aquaporin 4 (AQP4) being the main subtype expressed in the CNS. In the brain, AQP4 is expressed in astrocytes (*Papadopoulos and Verkman, 2013*; *Xiao et al., 2023*) that are a major type of glial cell in the parenchyma (*Barres, 2008*). This feature enables astrocytes to well balance osmotic oscillations imposed by the transmembrane transport of ions, metabolites, and signaling molecules, all mediated by water. AQP4 has been proposed to maintain the circulation of CSF, underlying the clearance efficacy of glymphatic system (*Hablitz et al., 2020*; *Kress et al., 2014*; *Mestre et al., 2018a*; *Smith et al., 2017*). Dysregulation in astrocyte aquaporin water transport is implicated in pathological scenarios such as cerebral edema and neuroinflammation (*Verkman et al., 2011*). Valuable insights have been obtained on the function of astrocyte aquaporin in specific physiopathological conditions including osmotically evoked volume responses, ischemia, and neurodegenerative diseases (*Verkman et al., 2017*). Meanwhile, how astrocyte aquaporin contributes to brain water homeostasis in basal physiology has remained elusive. A better understanding will aid to discern astrocytic regulation of cerebral water states, thereby the potential maladaptations in neurological disorders.

Here, combining in vivo chemical targeting, optical and MRI imaging of water dynamics, we report that AQP4 sustains a tonic water efflux from brain astrocytes. This mechanism was found to be critical in maintaining astrocyte volume and signaling stability, as well as the extracellular space remodeling in mouse cortex during optogenetically induced cortical spreading depression (CSD). Using DW-MRI, we observed that acute inhibition of astrocyte water efflux in vivo heterogeneously alters water diffusion across brain regions. Our finding suggests that aquaporin acts as a water export route in astrocytes in physiological conditions, contributing to maintain cellular and parenchymal water equilibrium.

## Results

### Astrocyte aquaporin mediates a tonic water efflux

To optically follow astrocyte water transport in situ, we performed real-time imaging of sulforhodamine B (SRB)-labeled astrocytes, whose fluorescence changes reflect real-time volume oscillation due to transmembrane water flux. We chemically labeled mouse brain astrocytes in vivo with the highly water-diffusible and astrocyte-specific red fluorescent dye SRB (*Figure 1A*, *top*) (*Appaix et al., 2012*; *Pham et al., 2020*). About 1 hr after its intraperitoneal injection, astrocytic SRB labeling was widely distributed in the mouse cortex, as seen in living acute brain slices (*Figure 1A*, *middle*) and validated by colocalization with EGFP-identified astrocytes in slices taken from GFAP-EGFP transgenic mice (*Figure 1A*, *bottom*, and *Figure 1—figure supplement 1*). Net water transport across the astrocyte membrane alters cytoplasmic SRB concentration and cellular volume, which can thus be followed by changes in fluorescence intensity when imaged in a fixed field of view. To validate this fluorescence intensity-translated (FIT) imaging of astrocyte water transport, we employed wide-field fluorescence microscopy for single-plane time-lapse imaging in acute brain slices in the primary somatosensory (S1) cortex. This approach allowed us to collect fluorescence from both the focal plane and along the axial extension, thereby imprinting volumetric fluorescence into the single image plane. Indeed, water influx induced by the application of hypoosmotic solution over different period decreased astrocyte SRB fluorescence, concomitantly reflecting the cell swelling (*Figure 1B*, *top*), whereas water export and astrocyte shrinking upon hyperosmotic manipulation increased astrocyte fluorescence (*Figure 1B*,

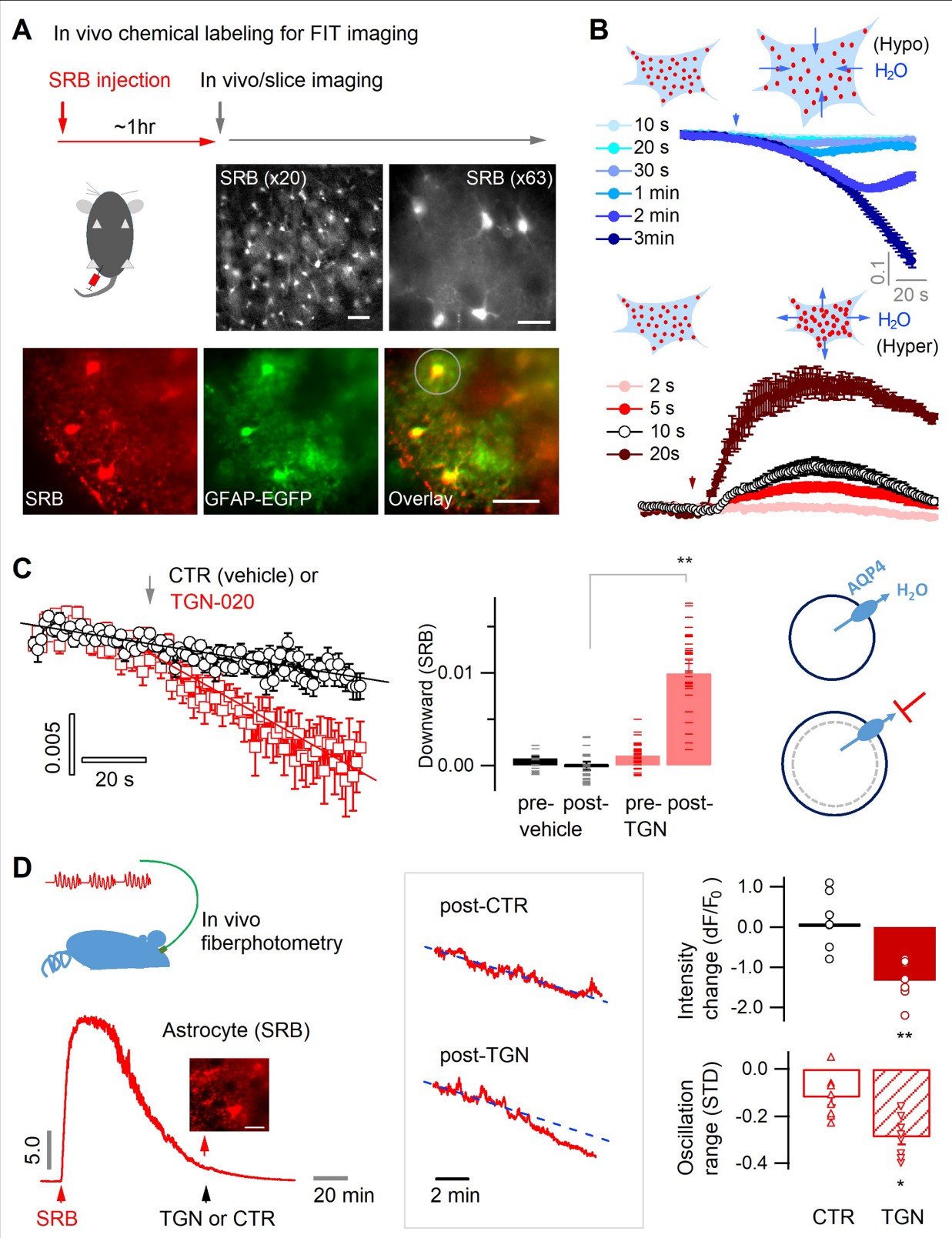

**Figure 1.** Astrocyte aquaporin mediates a tonic water efflux. (**A**) In vivo chemical labeling of astrocytes. Sulforhodamine B (SRB, 10 mg/mL) was intraperitoneally injected in awake mice (10 μL/g). Monochrome images show representative astrocyte labeling in living acute brain slices from cortex under low (×20; scale bar, 50 μm) and high magnification (×63; scale bar, 20 μm) by epifluorescence. *Below*, SRB labeling was confirmed to be astrocyte-specific in acute brain slices of the astrocyte reporter line GFAP-EGFP, where light sheet imaging was used to gain optical sectioning (Materials and

*Figure 1 continued on next page*

*Figure 1 continued*

methods; *Figure 1—figure supplement 1*). The light gray circle indicates the astrocyte regions used for fluorescence analysis. Scale bar, 50 μm. (**B**) Optical imaging of astrocyte water transport in acute brain slices. Transmembrane water transport was triggered with hypo- and hypertonic solution, inducing water inflow and outflow that were respectively reflected by SRB fluorescence decrease (*above*) and increase (*below*; expressed as dF/F$_0$). The hypo- or hypertonic solution was applied to slices over different lengths of duration displayed in colors corresponding to the time courses of SRB fluorescence (n=52 astrocytes, four mice). (**C**) Acutely blocking astrocyte AQP4 with TGN-020 caused intracellular water accumulation and swelling. The downward change in the SRB fluorescence was respectively calculated for the phases prior and post to vehicle or TGN application. *Left*, while no effect was seen under CTR condition (vehicle only, n=23 astrocytes, three mice), TGN-020 (20 μM) significantly decreased astrocyte SRB fluorescence (n=30, six mice). Imaging was performed in acute brain slices of layer II/III primary somatosensory (S1) cortex. *Middle*, the downward slope was compared between the periods before and after the application of TGN-020. *Right*, illustration shows astrocyte aquaporin sustaining a tonic water efflux. Its blockade causes water accumulation and cell swelling. (**D**) In vivo validation of the effect of TGN-020 application on astrocyte water homeostasis. *Left*, fiber photometry was used for real-time recording of SRB fluorescence of astrocyte population in S1 cortex in freely moving mice, with saline (CTR) or TGN-020 being intraperitoneally injected when SRB was trapped in astrocytes. Fiber photometry recording shows that in vivo SRB injection resulted in rapid entry into mouse cortex and, in about 1 hr, led to astrocyte labeling (inset scale bar, 50 μm). *Middle*, example response to saline and TGN-020. The change of SRB fluorescence relative to the photobleaching tendency delineated by line fitting (dotted line) was examined. *Right*, relative to CTR, TGN administration led to a decrease in astrocyte SRB fluorescence and its oscillation range (n=8 recordings per condition, five mice). Two-sample *t*-test was performed in Matlab, and error bars represent the standard errors.

The online version of this article includes the following figure supplement(s) for figure 1:

**Figure supplement 1.** Angular light sheet imaging in acute brain slices.

*bottom*). Hence, by following the fluorescence change of SRB-labeled astrocytes, the FIT imaging strategy allows to monitor the real-time changes in astrocyte water transport and volume change.

In basal conditions, flat fluorescence time course was recorded in brain slices suggesting equilibrated water transport across the astrocyte membrane (*Figure 1B*) and volume homeostasis in the brain parenchyma. To inspect a tonic role of aquaporin in astrocyte water transport, we sought to acutely block astrocytic aquaporin channel, AQP4. We used the synthetic compound 2-(nicotinamido)-1,3,4-thiadiazole (TGN-020) that is derived from the condensation of nicotinamide and thiadiazole derivatives (*Burnett et al., 2015*; *Huber et al., 2009*; *Igarashi et al., 2011*; *Sun et al., 2022*), whose specificity for AQP4 has been validated in vitro by ion channel heterologous expression system (*Toft-Bertelsen et al., 2021*) and in vivo using the AQP4 knockout (KO) mice (*Harrison et al., 2020*; *Igarashi et al., 2013*). This approach permitted the functional pinpointing of astrocyte aquaporin under physiological conditions, while avoiding the chronic compensations caused by genetic tools and mouse models that have been reported to alter brain water content, volume, and extracellular architecture potentially confounding the functional readouts (*Binder et al., 2004*; *Gomolka et al., 2023*; *Haj-Yasein et al., 2011*; *MacAulay, 2021*; *Yao et al., 2015*).

AQP4 is a bidirectional channel providing a path of least resistance for water transfer along the osmotic gradient (*Papadopoulos and Verkman, 2013*). As an osmotic equilibrium is maintained in the brain parenchyma in basal state, there might not be net water transport across AQP4. However, we observed in cortical slices that while little impact was seen with vehicle (CTR) on the tendency of fluorescence time courses of SRB-labeled astrocytes (*Figure 1C*, *left and middle*), acute inhibition of astrocyte AQP4 by TGN-020 caused a gradual decline, suggesting a progressive intracellular water accumulation (*Figure 1C*, *left and middle*). This observation implies that in basal condition, astrocyte aquaporin mediates a constant water efflux; its blocking causes intracellular water accumulation and swelling (*Figure 1C*, *right*). To corroborate this observation in vivo, we performed fiber photometry recording in moving mice (*Figure 1D*, *left*). An optical fiber was implanted into the mouse S1 cortex to follow fluorescence signals of local astrocyte population post the intraperitoneal injection of SRB (*Figure 1D*, *left*). After about 1 hr, when astrocytes were labeled, either saline or TGN-020 was injected intraperitoneally. Although no significant effect was observed with the saline control, TGN-020 induced a decrease in astrocyte SRB fluorescence and a reduction in its oscillation range (*Figure 1D*, *right*), mirroring an intracellular dilution of fluorescent molecules. Together, these observations suggest that acute inhibition of astrocyte aquaporin leads to intracellular water accumulation, thereby cell swelling.

## Aquaporin inhibition perturbs astrocyte and neuron signaling

Astrocyte volume equilibrium not only determines brain structural stability, but also associates with dynamic cellular signals. Astrocyte swelling has been shown to alter intracellular Ca$^{2+}$ signaling

(*Benfenati et al., 2011*; *Eilert-Olsen et al., 2019*). We performed Ca²⁺ imaging in acute brain slices to confirm that acute aquaporin inhibition induces astrocyte swelling and then Ca²⁺ oscillation. The genetically encoded Ca²⁺ sensor GCaMP6f was selectively expressed in astrocytes by crossing the Glast-Cre^ERT2 and GCaMP6^floxP mouse lines (*Herrera Moro Chao et al., 2022*; *Pham et al., 2020*; *Figure 2A*, *top*). Because astrocyte Ca²⁺ signals often occur in local domains, we adapted a light sheet microscope (*Pham et al., 2020*) for wide-field optical sectioning so as to image them with high signal-to-noise ratio (*Figure 1—figure supplement 1*). We first confirmed that water accumulation, therefore astrocyte swelling, induced by hypotonic solution enhanced the intracellular Ca²⁺ signal (*Figure 2A*). Then, we followed astrocyte Ca²⁺ in isotonic control condition while blocking astrocyte AQP4 with TGN-020, and observed that Ca²⁺ signaling was augmented following the acute aquaporin inhibition (*Figure 2B*). The spontaneous Ca²⁺ signals in perivascular astrocyte end feet, which wrapped the blood vessels displaying as tube-like structures, showed relatively higher sensitivity to the acute application of TGN-020, consistent with the enrichment of AQP4 in the end feet (*Figure 2—figure supplement 1*). This result confirms that inhibiting astrocyte tonic water efflux would have led to intracellular water accumulation and swelling, thus altering the astrocytic Ca²⁺ signaling. In addition, astrocyte swelling has been reported to induce the release of neuroactive molecules such as glutamate thereby influencing nearby neuron activity (*Fiacco et al., 2007*; *Yang et al., 2019*). Disrupting AQP4 tonic water outflow may not only cause astrocyte swelling but also influence the activity of neighboring neurons. We then imaged Ca²⁺ signals as a surrogate for neuronal activity in cortical somatostatin (Sst) interneurons which exhibit high excitability (*Karagiannis et al., 2021*) and express both ionotropic and metabotropic glutamate receptors (*Cauli et al., 2000*), rendering them ideal to sense astrocyte-released signaling molecules. We expressed GCaMP6f in Sst interneurons by crossing homozygous Sst-Cre (*Taniguchi et al., 2011*) and homozygous GCaMP6^floxP mice. TGN-020 inhibition of astrocyte AQP4 led to a global Ca²⁺ elevation in Sst interneuron populations (*Figure 2C*). These observations suggest that astrocyte aquaporin mediates a tonic water efflux, contributing to maintain both the volume and signaling homeostasis.

## Aquaporin water efflux regulates astrocyte volume response

We then examined the role of aquaporin water efflux in astrocyte volume response to osmotic changes. The AQP4 blocker TGN-020 or equimolar vehicle was applied throughout the recording, namely being continuously present before and after osmotic challenges. We first followed the evoked water efflux and shrinking induced by hypertonic solution. As astrocyte AQP4 supports preferentially water efflux, its inhibition would attenuate hypertonicity-imposed water extrusion (*Figure 3A*). Supporting this, application of TGN-020 slowed down the hypertonicity-induced water efflux and shrinking, reflected by the longer time to peak in the SRB fluorescence time course as compared to control (*Figure 3B*, *left*), though the delay in the initial onset was not significantly prolonged (*Figure 3B*, *right*). The maximal increase in astrocyte SRB fluorescence was lower in the presence of TGN-020, suggesting that AQP4 blocking reduced the overall amount of water efflux (*Figure 3B*, *right*).

We next evoked water influx, thereby astrocyte swelling, by hypotonic solution in the presence or absence of TGN-020 (*Figure 3C*). In contrast to the effects on hypertonicity-evoked water efflux, AQP4 acute inhibition was observed to accelerate both the initial water accumulation and the swelling rate in astrocytes, reflected by their earlier onset in SRB fluorescence decrease and faster reaching to the plateau value relative to the control condition (*Figure 3D*). This observation cross-validates the role of astrocyte aquaporin in supporting water efflux, whereby its blockade facilitates the initial accumulation of the evoked water influx (*Figure 3C*). In contrast, the late-stage maximum decrease in astrocyte SRB fluorescence was observed to be reduced in the presence of TGN-020 (*Figure 3D*, *right*), reflecting a reduction in the total amount of water accumulation and swelling. As TGN-020 was present prior to hypotonic challenge, it would have slightly swelled astrocytes due to the blockade of tonic water efflux, thereby constraining the range of further swelling induced by subsequent hypotonicity. In addition, while astrocyte AQP4 sustains a water outflow in basal physiology, in conditions when transmembrane water gradient is altered as occurred in hypotonic solutions, the net water flow through AQP4 should be finally dictated by the osmotic gradient due to its bidirectional nature. Therefore, along the imposed hypotonicity, AQP4 started to instruct a water influx; its constant blocking by TGN-020 would have reduced the total amount of water influx thereby the maximal extent of swelling.

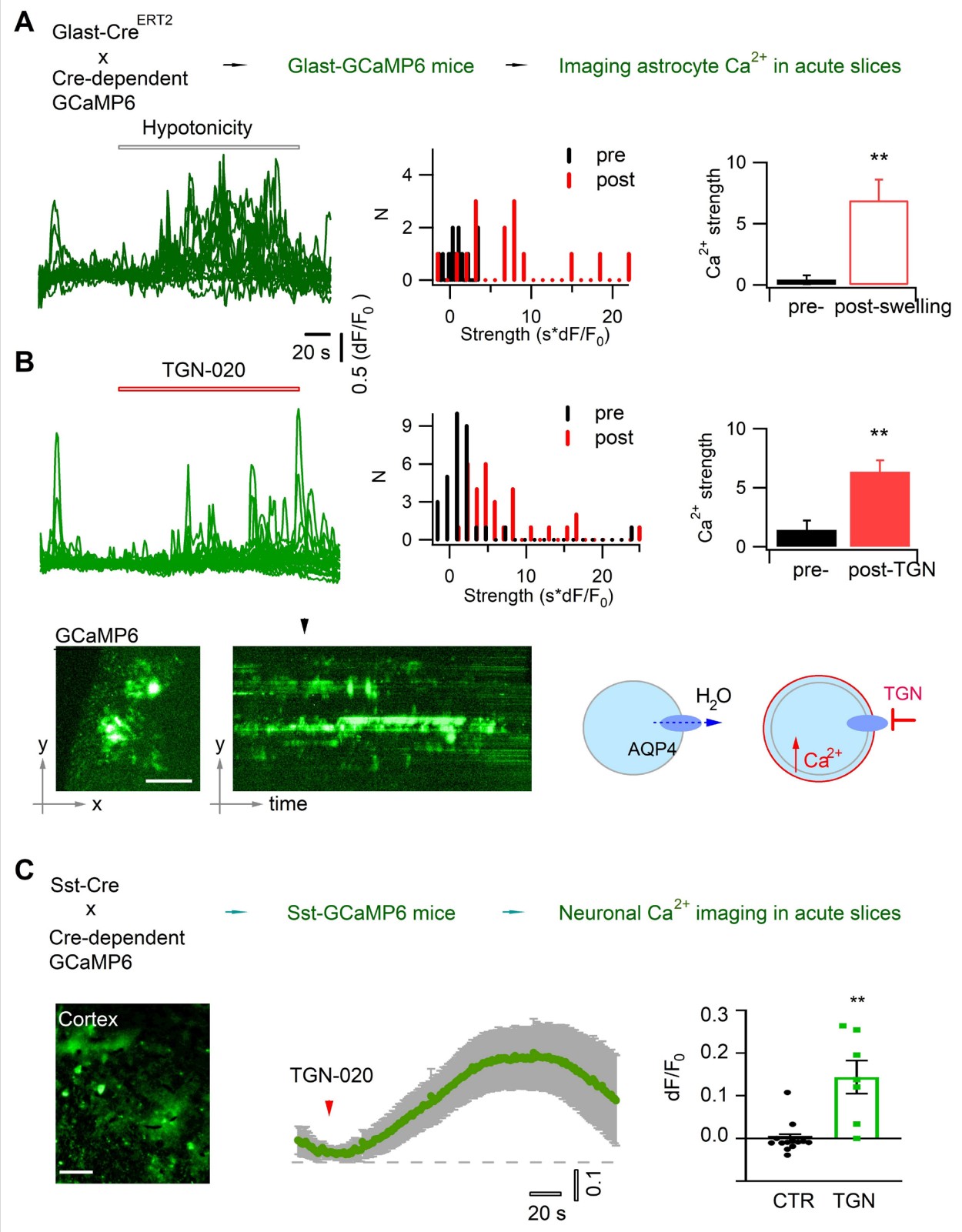

**Figure 2.** Acutely blocking astrocyte aquaporin induces swelling-associated Ca²⁺ oscillation. (**A**) In vivo expression in astrocytes of the genetically encoded fluorescent Ca²⁺ indicator GCaMP6 for imaging astrocyte Ca²⁺ in acute brain slices. Angular light sheet microscopy (***Figure 1—figure supplement 1***) was used to capture transient Ca²⁺ signals of local regions. As a positive control, astrocyte swelling was induced by hypotonic solution (100 mOsM) that caused Ca²⁺ changes from their homeostatic level. *Left*, representative time courses of swelling-induced Ca²⁺ changes in detected

*Figure 2 continued on next page*

*Figure 2 continued*

response regions; *middle to left*, the histogram distribution and bar representation showing the signal strengths that were derived from the temporal integral of individual Ca$^{2+}$ time courses normalized per minute, before and after cell swelling (n=15 regions of interest [ROIs], from three mice). (**B**) Astrocyte Ca$^{2+}$ oscillation caused by acute aquaporin blocking with TGN-020 (20 µM, n=31 ROIs, from five mice), due to the inhibition of the tonic water efflux that led to astrocyte swelling as illustrated. Time scale and Ca$^{2+}$ rise scale (dF/F$_0$) are the same as in (A). Scale bar, 50 µm. Mann-Whitney U non-parametric test was performed in Matlab (Wilcoxon rank sum test). (**C**) Intercellular effect on somatostatin (Sst) interneurons of blocking astrocyte aquaporin water efflux. TGN-020 (20 µM) or the equal molar vehicle (CTR) was bath applied to acute cortical slices of Sst-GCaMP6 mice (n=7–12 regional measurements; four mice). Mann-Whitney U non-parametric test was performed in the Prism software. Scale bar, 50 µm.

The online version of this article includes the following figure supplement(s) for figure 2:

**Figure supplement 1.** Calcium signals in perivascular astrocyte end feet.

Astrocytes are widely distributed throughout the brain parenchyma, orchestrating water transport and volume homeostasis (***Ochoa-de la Paz and Gulias-Cañizo, 2022***). Water outflow via astrocyte AQP4 may play a role in the structural remodeling of parenchyma at the global level. We then induced general cell swelling and changes in extracellular spaces by triggering CSD, a widespread depolarization implicated in brain physiopathology (***Chung et al., 2019***; ***Holthoff and Witte, 1996***; ***Zhao et al., 2019***). To be orthogonal to the pharmacological control of astrocyte aquaporin, the CSD depolarization wave was initiated by optogenetically stimulating ChR2-expressing glutamatergic pyramidal cells in the cortical slices of Emx1-Cre::Ai32ChR2 mice (***Gorski et al., 2002***; ***Madisen et al., 2012***), termed optogenetic cortical spreading depression (opto-CSD). As cell swelling increases the transmittance of infrared light, we imaged opto-CSD by the intrinsic optical signal (IOS) derived from infrared illumination (***Holthoff and Witte, 1996***; ***MacVicar and Hochman, 1991***), which was spectrally distinct from ChR2 photoactivation (detailed in Materials and methods). The IOS displayed transient increase across cortical layers during the opto-CSD (***Figure 4A–C***), exhibiting propagating kinetics characteristic of spreading depression (***Chung et al., 2019***; ***Holthoff and Witte, 1996***). The biphasic starting of IOS coincided with a sharp response in the extracellular potential indicating the CSD initiation (***Figure 4—figure supplement 1A***). IOS signal contains a first peak reflecting the rapid CSD response (***Figure 4C, b***) followed by a prolonged phase of general cellular swelling (***Figure 4C, c***). By combining fluorescence imaging of SRB-labeled astrocytes, whose spectrum is separated from the IOS infrared signal, we observed that astrocytes swelling (i.e. decrease in SRB fluorescence) paralleled CSD swelling (***Figure 4—figure supplement 2***). Consistent with our observation on astrocyte volume response (***Figure 3D***), when pre-incubating slices with TGN-020 to block AQP4 water outflow, the initiation of both the CSD and general swelling was accelerated while their maximum amplitude reduced (***Figure 4D and E***; ***Figure 4—figure supplement 1B***). Indeed, TGN-020 also increased the speed and the duration of the swelling while prolonging the recovery time from the swelling, confirming acute inhibition of AQP4 water efflux facilitates astrocyte swelling while restrains shrinking (***Figure 4—figure supplement 3***). Blocking action potentials with tetrodotoxin only reduced the amplitude of the initial CSD response while the effect on general swelling is insignificant (***Figure 4D and E***). We further followed the swelling of individual SRB-labeled astrocytes during opto-CSD. In consistence with the result obtained from hypotonic challenge (***Figure 3D***, *right*), the presence of TGN-020 reduced the peak amplitude of astrocyte swelling (i.e. the maximal SRB fluorescence decrease; ***Figure 4F***). As a post hoc adaptation to the transiently induced opto-CSD, astrocyte swelling was followed by a shrinking to recover cellular volume to the homeostatic level, which was recorded as a rebound in SRB fluorescence intensity. We noted that TGN-020 presence significantly restrained the recovery of SRB fluorescence back to the baseline (***Figure 4F***, *bottom*), showing that AQP4 inhibition hindered astrocyte water efflux, therefore the shrinking.

Collectively, our data show that astrocyte AQP4 sustains a tonic water outflow regulating the cellular volume response and the general cell swelling of the parenchyma.

## Tonic astrocyte water transport underlies brain homeostasis

Water homeostasis sets the basis for molecular diffusion in the brain, which instructs neurotransmitter availability, ion recycling, and metabolite trafficking. We then examined in vivo the role of astrocyte aquaporin outflow in brain water diffusion using DW-MRI (***Le Bihan and Iima, 2015***). It uses the diffusion of water molecules to generate contrast that can be quantified by the apparent diffusion coefficient (ADC) in the nerve tissue (***Beaulieu, 2002***; ***Le Bihan, 2014***). We used a 7 T MRI to image global

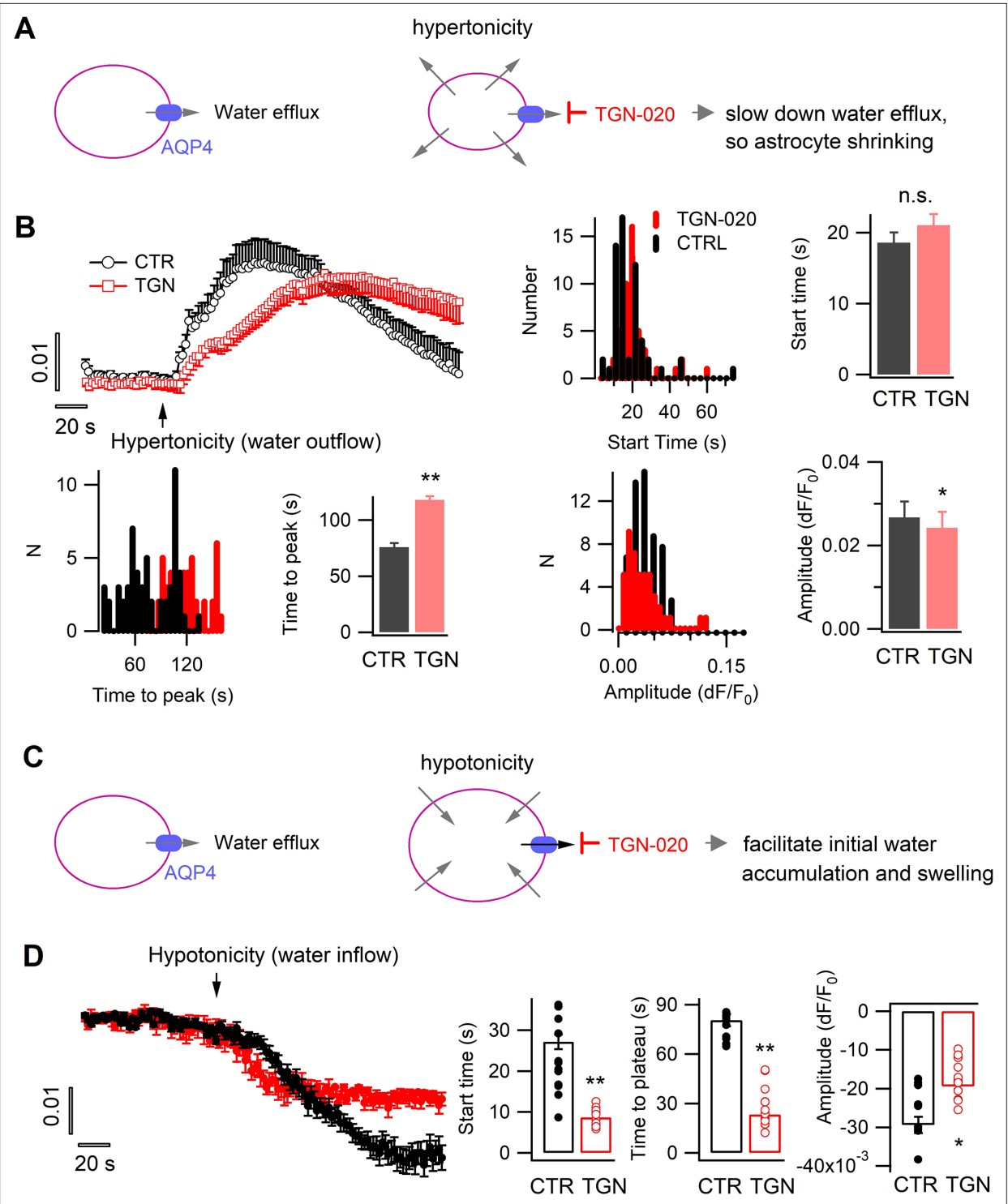

**Figure 3.** Tonic water efflux via aquaporin modulates phasic transmembrane water transport and astrocyte volume response. (**A**) *Left*, in basal condition astrocyte aquaporin mediates a tonic water efflux. *Right*, protocol to induce water outflow from astrocytes, therefore their shrinking, by hypertonic extracellular solution (400 mOsM) in either control condition (CTR) or in the presence of AQP4 inhibitor TGN-020 (20 µM). (**B**) Time courses of astrocyte sulforhodamine B (SRB) fluorescence increase upon the phasically induced water outflow, reflecting the occurrence of shrinking. The histograms and bar charts compare the start time, namely the delay between hypertonic solution application and rise in SRB fluorescence, the time to reach the peak of shrinking, and the absolute amplitude of water outflow-induced SRB increase (n=58 astrocytes for CTR, and 47 astrocytes for TGN-020, four mice). Mann-Whitney U non-parametric test was performed in Matlab (Wilcoxon rank sum test). (**C**) *Right*, protocol to trigger water inflow into astrocytes by hypotonic extracellular solution (100 mOsM) in either CTR solution or in the presence of TGN-020 (20 µM). (**D**) Time courses of astrocyte SRB

*Figure 3 continued on next page*

*Figure 3 continued*

fluorescence decrease caused by water inflow, which also reflects concomitant cell swelling. In contrast to the observation with hypertonicity-induced water outflow and astrocyte shrinking, a reduction was observed for both the start time and the time to peak with TGN-020 (n=12 astrocytes for CTR, and 12 astrocytes for TGN-020, four mice). TGN-020 led to a decrease in the absolute amplitude of astrocyte swelling.

brain water diffusion in mice lightly anesthetized with medetomidine that normalized the animals under the sedative state. Astrocyte AQP4 outflow was acutely perturbed by TGN-020 applied via intraperitoneal injection. Brain water diffusion was mapped every 5 min before and after TGN or saline (as control) administration (*Figure 5A*). Each acquisition sweep was performed with three b-values (0, 250, 1800 s/mm²) that accounted for ADC, so as to fit the exponential curve (*Figure 5B*). The representative images reveal the enough image quality to calculate the ADC, which allowed us to examine the effect of TGN-020 on water diffusion rate in multiple regions (*Figure 5C*). The temporal features in water diffusion change appeared also different: an early elevation followed by a reversible tendency was observed for cortical areas, a rapid and long-lasting elevation for the striatum while a delayed and transient increase for the hippocampus (*Figure 5D*, *top*). The respiratory rate was not changed by the injection of TGN-020 and saline (*Figure 5—figure supplement 1*). We assessed the time course of ADC change with three phantoms: distilled water, n-undecane, and n-dodecane. Both n-undecane and n-dodecane have been used as phantom for diffusion-weighted image (DWI) (*Tofts et al., 2000*; *Wu and Alexander, 2007*). The ADC was fluctuated within 2% of the averaged ADC at basal period (*Figure 5—figure supplement 2*), indicating that the ADC changes induced by TGN-020 is over the noise level. The in vivo neuroimaging results confirm that the tonic water efflux from astrocyte aquaporin contributes to maintain the homeostasis of brain water diffusion, and also suggest spatiotemporal heterogeneities in brain water handling.

## Discussion

Water equilibrium underlies brain function, plasticity, and dynamics. Astrocytes are the principal brain cell type expressing aquaporin water channels that are highly implicated in regulating local environments in the parenchyma (*Xiao et al., 2023*). We show that albeit being a bidirectional channel, astrocyte aquaporin sustains a tonic water outflow in basal states to ensure structural and functional stability, as well as the water homeostasis at the brain level. We have used the chemical dye SRB to label astrocytes in vivo, so as to follow the transmembrane water transport and volume dynamics by real-time fluorescence changes. This sensitivity is ensured by its far lower molecular weight (MW, 558.7 Da), thereby high solubility in the cytoplasm, as compared with fluorescence proteins such as the GFP (MW, 26.9 kDa). In addition, the chemical labeling enables genetics-less targeting of brain astrocytes, so as to avoid protein overexpression that may impact astrocyte volume and water transport responses.

The observation of a basal aquaporin water efflux implies there is a constitutive water accumulation in brain astrocytes. As a ubiquitous vehicle for transporting ions, transmitters, and metabolites, water enters astrocytes via a wide range of ion channels, transporters, and exchangers. For instance, they express $Na^+/K^+$ and $Na^+/HCO_3^-$ cotransporters to dynamically regulate intra- and extracellular ion homeostasis (*MacAulay, 2021*). Standing as a major glial cell type controlling the neuropil environment, astrocytes take up synaptically released $K^+$ through inwardly rectified $K^+$ channels and neurotransmitters (e.g. glutamate, GABA) via high-affinity transporters to safeguard synaptic transmission (*Dallérac et al., 2018*). The cotransport of water into the cells with ions or metabolites would lead to water buildup in astrocytes. Moreover, astrocytes juxtapose the cerebral vasculature, being a front relay station for brain metabolism. They express glucose transporters and lactate-permeable monocarboxylate transporters facilitating energy substrate uptake and transfer between blood vessels and neuron-glia networks (*Cauli et al., 2023*). As water is constantly produced during metabolic processes (*Bonvento and Bolaños, 2021*), astrocyte metabolism would also contribute to the constitutive water accumulation in the cytoplasm. An efficient efflux pathway appears necessary to counterbalance the constitutive water buildup, thereby maintaining astrocyte and brain homeostasis. Our data suggest that astrocyte AQP4 may fulfill such a role by mediating a tonic water outflow (*Figure 5—figure supplement 3*), providing functional evidence to the previous suggestion (*Amiry-Moghaddam et al., 2003*). It has been observed that KO of AQP4 leads to water overaccumulation in the brain, likely due

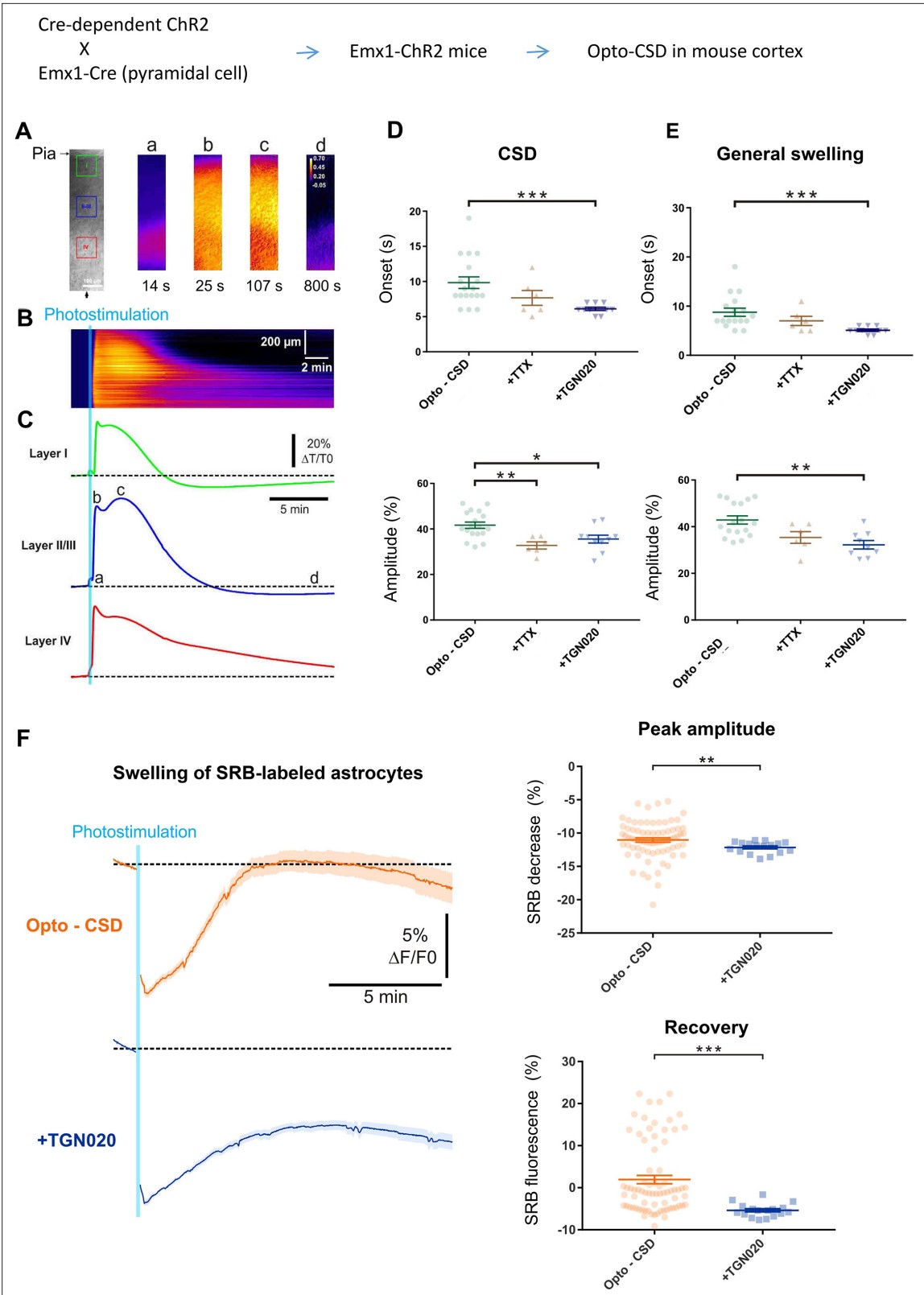

**Figure 4.** AQP4-mediated tonic water outflow regulates global swelling in cortical parenchyma. Cortical spreading depression (CSD)-associated general swelling was induced by photostimulating ChR2-expressing pyramidal cells in acute cortical slices, and recorded by imaging intrinsic optical signal (IOS) with infrared illumination. (**A**) Representative recording. The pia surface is on the upper side; green, blue, and red squares correspond to regions of interest in layers I, II–III, and IV, respectively. Transmittance signals are represented in pseudocolor images at different time points post photostimulation.

*Figure 4 continued on next page*

*Figure 4 continued*

Scale bar, 100 μm. (**B–C**) Kymograph and time courses showing the IOS changes following photoactivaiton, derived from the radial line of interest indicated by a black arrow in (A). The light blue line indicates the 10 s photostimulation that increases the infrared transmittance signal ($dT/T_0$) across cortical layers, as also illustrated by the time courses; a, b, c, and d correspond to the time points depicted in (A). After the onset delay (**a**), the first (**b**) and second (**c**) peak of IOS are characteristic of the CSD and a prolonged general swelling, respectively. (**D–E**) TGN-020 (20 μM) inhibition of AQP4 reduced the initial onset and the maximal amplitude of both the CSD and general swelling (n=6–18 measurements from 13 mice per condition) in layer II/III cortex. Inhibiting spiking activity with tetrodotoxin (TTX) (1 μM) only affected the amplitude of the initial CSD response. Bonferroni-Holm correction was used for multiplecomparisons. (**F**) Astrocytes swelling, reflected by sulforhodamine B (SRB) fluorescence decrease, monitored in control condition (n=75 astrocytes) and in the presence of TGN-020 (n=17 astrocytes) in layer II/III cortex. Mann-Whitney U non-parametric test was performed in Statsoft.

The online version of this article includes the following figure supplement(s) for figure 4:

**Figure supplement 1.** Optogenetic cortical spreading depression (opto-CSD) initiation and parameters analysis.

**Figure supplement 2.** Astrocytic swelling during an optogenetic cortical spreading depression (opto-CSD) across cortical layers.

**Figure supplement 3.** The swelling speed, duration of swelling, and recovery time during optogenetic cortical spreading depression (opto-CSD) (n=18 measurements from five mice for CTR, 10 measurements from four mice for TGN-020 for each parameter).

to the reduction in water exit from the pial membrane (*Haj-Yasein et al., 2011*; *Vindedal et al., 2016*). Our result also echoes that in the principal cells of the kidney collecting duct, AQP4 is suggested to mainly mediate water exit to balance the AQP2-sustained water entry (*Noda et al., 2010*). Moreover, we observed that inhibiting astrocyte basal water efflux alters $Ca^{2+}$ activities in both astrocytes and neurons, reminiscent of the early finding that inhibition of AQP4 by TGN-020 enhances cerebral blood flow in mouse cortex (*Igarashi et al., 2013*), which might be due to the enhanced neuron-astrocyte activity tone. As a pharmacological compound, TGN-020 exerts a partial blocking effect on AQP4 (*Huber et al., 2009*), implying that the actual functional impact of AQP4 per se might be stronger than what we currently observed. Alternatively, the phenylbenzamide AER-270 and its water-soluble prodrug AER-271 have been reported as AQP4 blocker (*Farr et al., 2019*) and impairs brain water diffusion and glymphatic fluid (*Giannetto et al., 2024*). AER270(271) has also been noted to be the inhibitor for κB nuclear factor whose inhibition could reduce CNS water content and influence brain fluid dynamics (e.g. the ADC of DW-MRI) in an AQP4-independent manner (*Giannetto et al., 2024*; *Salman et al., 2022*). The inhibition efficiency of AER-270 seems also lower than that of TGN-020 (*Huber et al., 2009*; *Giannetto et al., 2024*; *Salman et al., 2022*). Developing pharmacological compounds that can more specifically and efficiently target AQP4 will facilitate the functional and pre-clinical investigations.

This study also provides mechanistic hints to understand AQP4-relevant pathologies in specific contexts. For instance, astrocyte water accumulation and swelling are implicated in the development of brain edema. Different types of edema have been delineated depending on the pathological contexts, including cytotoxic, vasogenic, and hydrocephalic subtypes. The presence of AQP4 has been found to ameliorate vasogenic (vascular leak) and hydrocephalic edema (*Jeon et al., 2021*; *Tait et al., 2010*), where the excessive water has been suggested to exit the brain via AQP4-dependent route. This goes with our observation that astrocyte AQP4 contributes to basal water extrusion. Hence, aquaporin facilitators may help to control vasogenic and hydrocephalic edema. Differently, inhibiting AQP4 has been observed to ameliorate cytotoxic edema encountered during cerebral ischemia (*Verkman et al., 2011*). Beneficial effects of TGN-020 on edema have been reported in spinal cord injuries (*Li et al., 2019*), in ischemia (*Sun et al., 2022*) and in retinal edema in diabetic animal models (*Oosuka et al., 2020*). These observations suggest that the role of AQP4 in pathological conditions is context-dependent. Indeed, while our results suggest astrocyte AQP4 mediates a water outflow in basal states, the direction of water transport in specific contexts will be determined by the actual transmembrane water gradient. As the origin of edema is variable and water is among the many other ions involved in edema formation (*Stokum et al., 2016*), astrocyte aquaporin may either be the primary cause for cytosolic water accumulation or secondarily sustain protective water extrusion. In neuromyelitis optica spectrum disorders, an unusual nervous autoimmune disorder often featured with circulating IgG autoantibody against AQP4, the impairment in astrocyte water homeostasis has been observed (*Hinson et al., 2012*; *Lucchinetti et al., 2014*). The inhibition of astrocyte AQP4 by autoantibody may initially affect the basal water efflux. Yet chronic pathological evolution could convert astrocytes into reactive states, with either or both the aquaporin expression, intra- and extra-cellular environments being altered, the role of AQP4 in water transport might be accordingly shifted

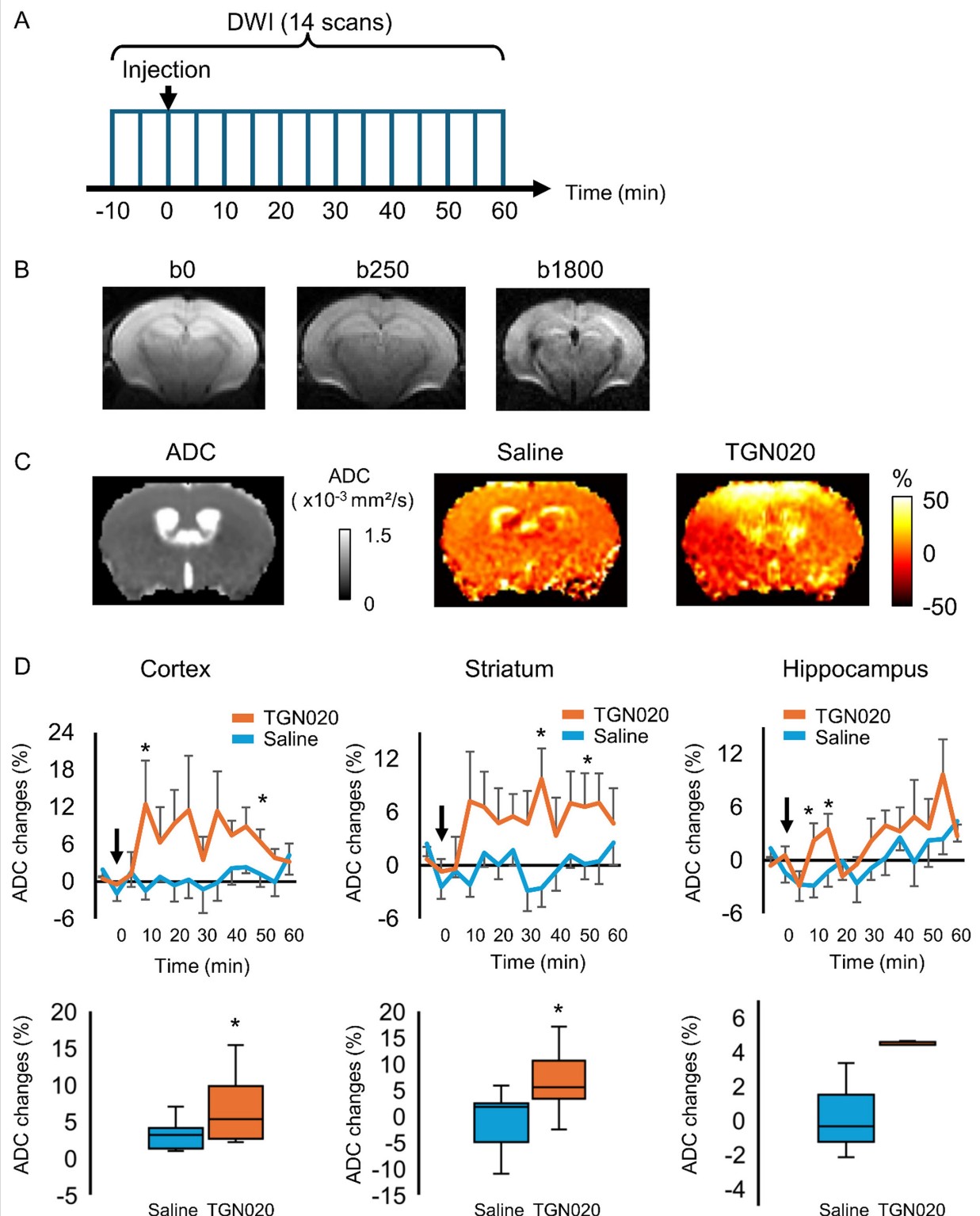

**Figure 5.** Perturbing the tonic water efflux via astrocyte aquaporin alters brain water homeostasis. In vivo diffusion-weighted magnetic resonance imaging (DW-MRI) (7 T) was employed to map water diffusion in the entire brain scale following the acute inhibition of astrocytic AQP4 (TGN-020, intraperitoneal injection, 200 mg/kg), with paralleled control experiments performed with saline injection. (**A**) Experimental protocol for DW-MRI. Saline or TGN-020 was injected at 10 min after the start of acquisition. Diffusion-weighted image (DWI) was acquired every 5 min. (**B**) Representative image obtained at three different b-values to derive the water diffusion rates. (**C**) *Left*, brain water diffusion rate was mapped by the calculation of apparent diffusion coefficient (ADC). *Right*, representative images illustrating the relative changes of ADC at 60 min after injection of saline or TGN-020. (**D**) *Upper*

*Figure 5 continued*

*panel*, time courses depicting the temporal changes in ADC in the cortex, striatum, and hippocampus, revealing the regional heterogeneity. Arrowhead indicates the injection of saline or TGN-020 (n=7 mice for saline injection, 7 mice for TGN-020). *p<0.05 vs baseline at each time point following the two-way repeated measures ANOVA. *Lower panel*, box-plot of the averaged ADC between 30 and 50 min after the injection. *p<0.05 by paired t-test.

The online version of this article includes the following figure supplement(s) for figure 5:

**Figure supplement 1.** Box-car plot of averaged respiratory rate before and after injection of saline or TGN-020.

**Figure supplement 2.** Apparent diffusion coefficient (ADC) in water, undecane, dodecane, and fluorine (proton-free liquid).

**Figure supplement 3.** The present observations suggest that the tonic water efflux via AQP4 helps to counterbalance excessive water accumulation in astrocytes, thereby maintaining water, volume, and signaling homeostasis in the brain.

(*Mireles-Ramírez et al., 2024*). Additionally, pro-inflammatory cytokines and chemokines during the immunological maladaptation could also contribute to shape astrocyte functions including aquaporin-mediated water transport (*Lucchinetti et al., 2014*; *Mireles-Ramírez et al., 2024*). Recent data show that TGN-020 targeting of AQP4 mitigates the inflammation during ischemia-reperfusion injury, by enhancing glymphatic amyloid clearance and inhibiting the ERK 1/2 pathway (*Li et al., 2024*), suggesting that in pathological conditions, multiple factors could be recruited by astrocyte aquaporin targeting.

We observed that acutely blocking aquaporin caused basal water accumulation leading to swelling in astrocytes, and facilitated the initial occurrence of the evoked swelling by hypoosmoticity and during CSD. This recalls early observation by electron microscopy that knocking out the anchoring protein alpha-syntrophin to disperse AQP4 clustering induces an enlargement of astrocyte end feet (*Amiry-Moghaddam et al., 2003*). It likely reflects astrocyte local swelling caused by the disruption of AQP4-mediated water efflux. Our observation also parallels the report that AQP4 KO facilitates astrocytes swelling induced by hypoosmotic solution (*Murphy et al., 2017*). Nevertheless, genetic inactivation of AQP4 also reveals inhibitory effects on astrocyte swelling (*Benfenati et al., 2011*; *Mola et al., 2016*; *Woo et al., 2018*). Such discrepancy might be due to the variable compensations in brain water content and structure integrity during chronic genetic manipulations (*Binder et al., 2004*; *Haj-Yasein et al., 2011*; *MacAulay, 2021*; *Yao et al., 2015*). Using the calcein quenching assay and AQP4 KO (*Solenov et al., 2004*), it has been demonstrated in cultured astroctyes that AQP4 is a functional water channel. AQP4 deletion reduced both astrocyte water permeability and the absolute amplitude of swelling over comparable time, and also slowed down cell shrinking, which are overall confirmed by in our results from acute AQP4 blocking. Yet in the study (*Solenov et al., 2004*), the time to swelling plateau was prolonged in AQP4 KO astrocytes, which was observed to be accelerated in the present study. This difference may be due to compensatory mechanisms in chronic AQP4 KO, or reflect the different volume responses in cultured astrocytes from brain slices or in vivo contexts as noted previously (*Risher et al., 2009*). Nevertheless, in the present study, astrocyte volume and water transport dynamics have been inferred from the real-time fluorescence change of SRB labeling. Rapid three-dimensional imaging of astrocyte volume dynamics in situ will provide further quantitative information on the role of AQP4 in volume regulations (*MacAulay, 2021*; *MacAulay et al., 2004*). In addition, though our data suggest that under basal condition astrocyte AQP4 sustains a tonic water outflow, in conditions when massive water influx persists as exemplified by the application of hypoosmotic solutions, the direction of water transport of AQP4 would follow the imposed water gradient. Regulatory volume decrease has been observed in cultured astrocyte volume responses behaving as a protective mechanism (*Mola et al., 2016*). In current condition in acute brain slices, when we applied hypotonic solution to induce astrocyte swelling, our protocol did not reveal rapid regulatory volume decrease. During opto-CSD, we observed astrocyte volume decrease following the transient swelling (*Figure 4F*), where both the peak amplitude and the degree of recovery were reduced by inhibiting AQP4 with TGN-020. These observations suggest astrocyte regulatory volume decrease is likely to be condition-dependent, while also recalling that the regulatory volume decrease is barely detectable in brain slices or in vivo during hypoosmotic challenge (*Risher et al., 2009*). Given the implication of CSD in brain migraine aura, ischemia, and seizure (*Chung et al., 2019*), the current observation suggests a functional involvement of astrocyte AQP4 in the physiopathological adaptations.

Astrocyte aquaporin modulates brain water transport (*MacAulay, 2021*), though its role in the glymphatic system is under deliberation (*Mestre et al., 2018a*; *Rasmussen et al., 2022*; *Smith and*

*Verkman, 2019*; *Smith et al., 2017*). Our data suggest that astrocyte aquaporin-mediated outflow helps to maintain local water environment in the brain parenchyma, facilitating the circulation of the interstitial fluid and CSF. Moreover, besides affecting water transport, targeting AQP4 also impacts astrocyte volume, therefore the size of extracellular space. These two factors would need to be considered when evaluating AQP4 involvement in brain fluid transport.

We show by DW-MRI that impairing astrocyte water transport induces the changes of ADC across the brain. When blocking AQP4 with TGN-020, therefore its basal water outflow, we observed spatially heterogeneous elevations in brain water diffusion rate. It has been reported that relative to wild-type mice, AQP4 KO mice show increased water diffusion along with the enlarged interstitial space, brain volume yet reduced CSF spaces (*Gomolka et al., 2023*). Notably in AQP4 KO mice, the influx of gadolinium CSF tracer was reduced as followed by dynamic contrast-enhanced MRI (*Gomolka et al., 2023*). As corroboration, the AQP4 blocker AER-271 also reduces glymphatic water influx (*Giannetto et al., 2024*). Together, these observations suggest that AQP4 plays a critical role not only in modulating water diffusion in brain parenchyma but also in the CSF transport. Our current data are derived from 5 min apart acquisitions, providing information over the early phase of AQP4 inhibition, therefore extending our early report (*Debacker et al., 2020*). The regional heterogeneity likely reflects the various levels of AQP4 expression; its relative enrichment in the cerebral cortex (*Gomolka et al., 2023*; *Hubbard et al., 2015*; *Mestre et al., 2018a*) corresponds to the pronounced effect observed here by DW-MRI. An overall increase in brain water diffusion rate was observed when blocking AQP4 water efflux. This treatment causes water accumulation inside astrocytes and their swelling, which reduces the extracellular space but increases the intracellular space. Water diffusion would be enhanced inside astrocytes and decreased in extracellular space, respectively. DW-MRI maps global water diffusion of both intra- and extracellular space. Likely, a net increase in brain water diffusion may reflect its intracellular increase exceeding the extracellular decrease. In addition, convective brain fluid flow has been suggested to be present in the perivascular space (*Smith and Verkman, 2019*), a scenario extended by the glymphatic system to the extracellular space of the parenchyma (*Mestre et al., 2018b*). In this sense, transiently squeezing extracellular space might increase the rate of the water flow. Additionally, a transient upregulation of other water-permeable pathways would increase the water diffusion in compensating the AQP4 blocking. It has been shown that TGN-020 increases regional cerebral blood flow in wild-type mice but not in AQP4 KO mice (*Igarashi et al., 2013*). The current MRI data yet lack sufficient spatial resolution to delineate the spatial compartmentalization of brain fluid flow. We here used anisotropic resolution and minimum b-values to earn the temporal resolution. The phantom study supports the quality of DW-MRI data to calculate the time course of ADC. To further use isotropic resolution and more b-values will help to validate our current results.

This study sheds light on the mechanisms by which astrocyte aquaporin contributes to the water environment in brain parenchyma, and will help to understand the processes underlying brain homeostasis and adaptation to life conditions.

## Materials and methods
### Animals and acute brain slice preparation

All procedures using animals were carried out in strict accordance with French regulations (Rural Code R214/87 to R214/130) and conformed to the ethical recommendations of the European Economic Community (Directive 86/609/EEC) and the National Charter French on ethics in animal experimentation. All protocols were approved by the Charles Darwin ethics committee and submitted to the French Ministry of Education and Research (Approval 2015 061011367540 APAFIS#573-2015061011367547 v1).

For water transport and volume imaging, coronal slices comprising somatosensory cortex were acutely prepared from C57BL/6 mice of both sexes at ages of 4–6 weeks, unless otherwise indicated. Mice were deeply anesthetized by isoflurane (ISOVET, Piramal) evaporation in a closed plexiglass box. The brain was taken out and placed in a modified artificial cerebrospinal fluid (aCSF) for slicing (in mM: 30 NaCl, 4.5 KCl, 1.2 NaH$_2$PO$_4$, 1 MgCl$_2$, 26 NaHCO$_3$, 10 D-glucose, and 194 sucrose) maintained at 4°C during sectioning, where depolarizing ions (Na$^+$, Ca$^{2+}$) were reduced in attempt to lower the potential excitotoxicity during the tissue dissection while sucrose used to balance the osmolarity (*Jiang et al., 2016*). The brain was cut into 300-µm-thick slices with a vibratome (Leica VT1200S).

Brain slices were recovered in standard aCSF (mM: 124 NaCl, 4.5 KCl, 1.2 NaH$_2$PO4, 1 MgCl$_2$, 2 CaCl$_2$, 26 NaHCO$_3$, and 10 D-glucose) at 37°C for about 1 hr, and the same aCSF was used for brain slice imaging at room temperature. For optogenetic cortical spreading depression (opto-CSD) experiments, 18- to 21-day-old mice were deeply anesthetized with 150 µL of isoflurane. After the decapitation, the brain was quickly removed and placed into ice-cold oxygenated aCSF containing the following (in mM): NaCl 125; NaHCO$_3$ 26; sucrose 15; glucose 10; KCl 2.5; CaCl$_2$ 2; NaH$_2$PO$_4$ 1.25; MgCl$_2$ 1; kynurenic acid 1 (nonspecific glutamate receptor antagonist, Sigma-Aldrich). Coronal slices from mouse somatosensory cortex (300 µm thick) were cut with a vibratome (VT1000, Leica) and allowed to recover at room temperature for at least 45 min in aCSF saturated with O$_2$/CO$_2$ (95%/5%).

## In vivo chemical and genetic targeting of astrocytes

Mice of similar age were also used for in vivo labeling of astrocytes, where the astrocyte-specific red fluorescent dye SRB (10 mg/mL, Sigma) was intraperitoneally injected into awake mice at a dose of 10 µL/g. The genetically encoded Ca$^{2+}$ indicator GCaMP6 was expressed in astrocytes in vivo by crossing a Cre-dependent GCaMP6f mouse line (Ai95-D, B6J.Cg-Gt(ROSA)26Sor$^{tm95.1(CAG-GCaMP6f)Hze}$/MwarJ, The Jackson Laboratory, Strain #028865) with an inducible conditional mouse line Glast-Cre$^{ERT2}$, which expresses Cre recombinase selectively in astrocytes upon tamoxifen injection (*Slezak et al., 2007*; MGI:4420274). Tamoxifen (T5648, Sigma) was dissolved in 100% ethanol (10 mg/250 µL). The corn oil (Sigma) was subsequently added at a 9:1 proportion relative to the ethanol-tamoxifen solution. The resultant solution was then heated at 37°C for 15 min, vortexed, sonicated for 15 min to reach transparency, before being aliquoted at 250 µL each containing 1 mg tamoxifen. Aliquots were stored at –20°C until use. Glast-Cre$^{ERT2}$::Ai95$^{GCaMP6f/WT}$ mice were injected with tamoxifen at ~3–4 weeks of age, once a day (1 mg) for 2 consecutive days. Genotyping of the Cre-dependent GCaMP6f mouse line used the standard primers and polymerase chain reaction protocols provided by The Jackson Laboratory. Genotyping of Glast-Cre$^{ERT2}$ mouse line was performed using the primers for Cre recombinase (TK139/141) as reported (*Slezak et al., 2007*). For Ca$^{2+}$ imaging in Sst interneurons, homozygous mice B6J.Cg-Sst$^{tm2.1(cre)Zjh}$/MwarJ (The Jackson Laboratory, Strain #028864) (*Taniguchi et al., 2011*) have been crossed with homozygous Ai95$^{GCaMP6f/GCaMP6f}$ mice to obtain heterozygous Sst-Cre$^{cre/WT}$::Ai95$^{GCaMP6f/WT}$ mice. For opto-CSD, homozygous mice B6.129S2-Emx1$^{tm1(cre)Krj}$/J (*Gorski et al., 2002*, Emx1-Cre$^{cre/cre}$, The Jackson Laboratory, Strain #005628) have been crossed with homozygous mice B6;129S-Gt(ROSA)26Sor$^{tm32.1(CAG-COP4*H134R/EYFP)Hze}$/J (*Madisen et al., 2012*, Ai32$^{ChR2/ChR2}$, The Jackson Laboratory, Strain #012569) to obtain heterozygous Emx1-cre$^{cre/WT}$::Ai32$^{ChR2/WT}$ mice.

## Ex vivo fluorescence imaging

Light sheet and epifluorescence imaging were both performed using a wide-field upright microscope (Zeiss Axioskop 50, Germany) equipped with water immersion objectives. Epifluorescence illumination was provided by a monochromator light source (Polychrome II, TILL Photonics, Germany) directly coupled to the imaging objective via an optical fiber. We used water immersion imaging objectives (ZEISS): ×10 NA0.3 used to have a global view, ×20 NA0.5 used to perform light sheet imaging, and ×63 NA1.0 to image individual astrocytes under epifluorescence configuration so as to collect their whole-cell SRB fluorescence to follow astrocyte water transport and volume dynamics. The green fluorescence of GCaMP6 was separated from the red fluorescence of SRB, by spectrally exclusive double-band filters (Di03-R488/561-t3 and FF01- 523/610, Semrock). SRB-labeled astrocytes were excited at 550 nm. Fluorescence signal was collected using a digital electron-multiplying charge-coupled device (EMCCD Cascade 512B, Photometrics).

A light sheet imaging system was adapted from our previous publication (*Pham et al., 2020*), where a chamber with a slightly tilted surface was used so as to spare the agarose embedding and allow the light sheet to access to any parts of the brain slice (*Figure 1—figure supplement 1*). Briefly, the in-focus light sheet was introduced laterally from an independent optical module (Alpha3 light sheet add-on, Phaseview, France) equipped with an air objective (Zeiss EC EPIPlan ×10, 0.25NA). This optical module was orthogonal to the upright imaging pathway and coupled via a wavelength combiner (Thorlabs) to two continuous wave lasers 473 nm and 561 nm (CNI, China), for the excitation of GCaMP6 and SRB, respectively. The imaging chamber (20×30×46 mm$^3$ width, height, and length) was custom-designed using 3D modeling software (Trimble SketchUP), produced by a Mojo 3D printer (Stratasys) with the acrylonitrile butadiene styrene material, where the horizontal slice-supporting

surface in the chamber was tilted by 8° along the laser entry direction. In this way, the brain slice could be laid on the slightly tilted surface while being stabilized with a conventional Harp slice anchor grid (Warner Instruments), whereby the laterally generated laser light sheet could reach local regions throughout the brain slices. The chamber was mounted on a motorized stage (PI-Physik Instrumente GmbH, Germany) for axial micro-manipulation. Laser excitation and image acquisition were controlled by MetaMorph (Molecular Devices).

Based on the time-lapse image stacks (1 Hz), the active regions of interest (ROIs) of astrocytes displaying dynamic $Ca^{2+}$ signals were identified using our established spatiotemporal screening method (*Pham et al., 2020*). The signal strength of astrocyte $Ca^{2+}$ signals was estimated from the temporal integral of the normalized $Ca^{2+}$ time courses (dF/$F_0$, $F_0$ is the average of the lowest 20-data point baseline) for the periods prior and post to TGN or vehicle application respectively, which was then normalized per minute. As for neuronal $Ca^{2+}$ signals, we followed the global changes with low-frequency imaging (1 Hz) and calculate the mean fluorescence of single 200×200 μm$^2$ field of view. SRB fluorescence dynamics of individual astrocytes was analyzed with ImageJ software (NIH). We sought to follow the temporal changes in SRB fluorescence signal. The acquired fluorescent images contained not only the SRB signal, but also the background signals consisting of, for instance, the biological tissue auto fluorescence, digital camera background noise, and the leak light sources from environments. The value of the background signal was estimated by the mean fluorescence of peripheral cell-free subregions (15×15 μm$^2$) and removed from all frames of single time-lapse image stacks. $F_0$ was identified as the mean value of the 10 data points immediately preceding the detected fluorescence changes, and the same timing used when comparing the paired conditions. Because single astrocytes typically occupy an area of ~50–100 μm in radius (*Appaix et al., 2012*), ROIs of 50 μm diameter were delineated surrounding identified astrocyte footprint, to comprise the extended area of individual astrocytes while minimizing interference with neighbors. Time courses extracted from single ROIs were normalized as dF/$F_0$.

## Dynamic imaging of the extracellular space and astrocytes volume changes

Individual slices were transferred to a submerged recording chamber and perfused with oxygenated aCSF (~2 mL/min) lacking kynurenic acid. Slices were observed using a double port upright microscope (BX51WI, Olympus, Japan) equipped with Dodt gradient contrast optics (DGC; Luigs and Neumann), collimated light emitting device (LED; 780 nm, Thorlabs) as transmitted light sources, ×4 objective (PlanN, 0.10NA, Olympus), ×20 (UMPlanFL N, 0.50 W, Olympus) or ×40 (LUMPlanF/IR, 0.80 W, Olympus) objectives, and a digital camera (OrcaFlash 4.0, Hamamatsu) attached on the front port of the microscope. The observation with ×4 allows to delimit the cortical layers and to confirm the presence of 'barrels'-like patterns in layer IV (*Erzurumlu and Gaspar, 2020*). The ×20 objective was used to observe infrared IOS (*Holthoff and Witte, 1996*; *MacVicar and Hochman, 1991*) of the cortical layers I–IV. The ×40 objective was used to image SRB-labeled astrocytes combined with IOS in layers I and II/III.

After 1 min of baseline period, a 10 s continuous photostimulation at 470 nm of the ChR2$_{H134R}$ (30% of the maximal power of the LED) was carried out via the epifluorescence port of the microscope using a collimated LED system (CoolLED, PreciseExcite) and a set of multiband filters consisting of an excitation filter (HC 392/474/554/635, Semrock), a dichroic mirror (BS 409/493/573/652, Semrock) and an emission filter (HC 432/515/595/730, Semrock) to observe the induced changes of light transmittance. The infrared and epifluorescence images were acquired each at 1 Hz (exposure 100 ms, binning 2×2) using the Imaging Workbench 6.1 software (INDEC BioSystems) for a total duration of 15 min.

## In vivo fiber photometry

The optical fiber (200 μm core, NA = 0.37, Neurophotometrics) coupled to a stainless-steel ferrule (1.25 mm) was stereotaxically implanted in mouse S1 cortex (relative to bregma, rostro-caudal 0 mm, lateral ± 2.5 mm, vertical –1.2 mm), and fixed to the skull with dental cement (SuperBond, Sun Medical). Mice were familiarized with the setup for 30 min prior to fiber photometry recording in mobile states. 5 min baseline was recorded before SRB injection, then SRB was applied to label brain astrocytes via intraperitoneal injection and the fluorescence time course was continuously recorded. To examine in vivo the effect on astrocyte water transport of acute AQP4 blocking, either TGN-020 (200 mg/kg;

Tocris or MedChemExpress) or saline control was intraperitoneally injected (*Debacker et al., 2020*), 1 hr after SRB injection when astrocytes were labeled as confirmed by brain slice imaging.

Fluorescence signals were recorded using a Neurophotometrics fiber photometry system (FP3002). A branched fiber-optic patch cord (BFP_200/230/900-0.37_FC-MF1.25, Doric Lenses) connected to the fiber photometry apparatus was attached to the implanted fiber-optic cannula using a cubic zirconia sleeve. To record fluorescence signals from SRB, excitation light from 560 nm LED was band-pass filtered, collimated, reflected by a dichroic mirror and focused by a ×20 objective. The excitation light power at the tip of the patch cord was 50 μW. Emitted fluorescence was band-pass filtered and focused on digital camera. Signals were collected at a rate of 1 Hz for SRB and visualized using the open-source software Bonsai 2.4 (http://bonsai-rx.org) and analyzed in Igor software (WaveMetrics).

## In vivo DW-MRI

To test the effect of astrocyte AQP4 acute inhibition on brain water homeostasis, we used DW-MRI to calculate the ADC of tissue water molecules in the brain. MRI experiment was performed with medetomidine anesthesia (0.6 mg/kg/hr, i.v.). Two groups of mice (seven for each) were injected with either the water-soluble TGN-020 sodium salt (Key Organics, LR-0041) or saline. For the crossover test, the order of injection of TGN-020 or saline was randomly determined. The experiment was performed 2 days, which separated 1 month for the recover. The respiration was monitored, and the body temperature was maintained at 36°C throughout the measurement.

The MRI experiments were conducted on a 7.0 T MRI system equipped with actively shielded gradients at a maximum strength of 700 mT/m (Biospec; 70/16 Bruker BioSpin, Ettlingen, Germany) and with a cryogenic quadrature radio frequency surface probe (Cryoprobe). We chose the cryoprobe because the signal-to-noise ratio appeared better than volume coil. We also confirmed to obtain good image quality in whole brain (*Figure 5B and C*). After 10 min from the start of the scanning, either TGN-020 (200 mg/kg body weight) or saline was injected. Then, scanning was continued for 60 min. The respiration rate was monitored and was confirmed within the range of 80–150/min throughout the experiment. Mice were injected with 200 mg/kg of TGN-020. We used a standard diffusion-weighted spin echo EPI sequence, with the following parameters: spatial resolution = 175×175×500 μm$^3$/voxel, three b-values (b=0, 250 and 1800 s/mm$^2$), six directions, one segment, echo time = 37.1 ms/repetition time = 5769 ms, bandwidth = 300 kHz, δ=3 ms, Δ=24 ms, scan time = 5 min. The DWI scan was continued for 70 min (total 14 scans). We used minimum direction of motion probing gradient and anisotropic resolution to image the time course of ADC changes following the injection of the TGN-020 (*Debacker et al., 2020*). Previous study has used long diffusion time (>20 ms) and long echo time (40 ms) to follow the mean diffusivity (*Aggarwal et al., 2020*), supporting the suitability of our protocol to investigate the ADC. The stability of DW-MRI was evaluated through 70 min scans using three different phantoms: water, n-undecane (Merck, Darmstadt, Germany), and n-dodecane (Merck, Darmstadt, Germany).

The motion correction of DWI data was performed using statistical parametric mapping software (SPM12, Welcome Trust Center for Neuroimaging, UK). The template image and ROIs were then co-registered to the DWI data using SPM12. The ROIs of the cortex, the hippocampus, and the striatum were depicted with reference to Allen Mouse Brain Atlas (https://scalablebrainatlas.incf.org/mouse/ABA12). The ADC at each time point was calculated from DWI data with all b-values (b=0, 250, and 1800 s/mm²) using DSI Studio (https://dsi-studio.labsolver.org/). The percentage change in ADC was then calculated according to the equation: $\Delta ADC_i = (ADC_i/ADC_2 - 1) \times 100$ (%). The $ADC_i$ is the ADC at the ith time point. We define the averaged ADC at first and second time point as the baseline because saline injection was performed after the second DWI scan. The averaged ADC changes within ROIs were calculated using a homemade program.

## Statistics

Experimental data are expressed as mean ± standard error unless otherwise mentioned. The t-test was performed for two-group comparisons and significant difference was determined by p-values less than 0.05. Mann-Whitney U test was used when the data were non-parametric deviating from normal distributions. Bonferroni-Holm correction was used for multiple comparisons. Statistical tests were carried out with MATLAB (The MathWorks) and Statistica 6 (StatSoft). The significance levels are shown in figures by *p<0.05, **p<0.01, ***p<0.001.

## Acknowledgements

We thank the animal and imaging facilities of the IBPS (Sorbonne Université, Paris, France). We thank Frank W Pfrieger for the Glast-Cre^ERT2 mice, Nathalie Rouach for the GFAP-EGFP mice, Hervé Le Corronc and Thomas Panier for the discussion and the 3D printing of the light sheet chamber, respectively. This work was supported by the Agence Nationale de la Recherche (ANR-17-CE37-0010-03; ANR-20-CE14-0025-02), Japanese Society for Promotion of Science (JSPS) 2022 summer grant, Grant-in-Aid for Challenging Research (Exploratory) in Japan (grant number 21K19464), the i-Bio initiative grant and the Emergence program of Sorbonne Université, and the grant from the Fondation de France. CP was supported by a fellowship from France Alzheimer and BLG by a fellowship from the Fondation pour la Recherche sur Alzheimer.

## Additional information

### Funding

| Funder | Grant reference number | Author |
| --- | --- | --- |
| Agence Nationale de la Recherche | ANR-17-CE37-0010-03 | Bruno Cauli |
| Agence Nationale de la Recherche | ANR-20-CE14-0025-02 | Dongdong Li |
| Grant-in-Aid for Challenging Research (Exploratory) in Japan | 21K19464 | Tomokazu Tsurugizawa |
| Japan Society for the Promotion of Science | Summer program 2022 | Cuong Pham |
| Sorbonne Université | i-Bio | Bruno Cauli |
| Sorbonne Université | Emergence | Dongdong Li |
| Fondation de France | 00135008 / WB-2022-4423 | Dongdong Li |
| France Alzheimer et maladies apparentées | PhD fellowship | Cuong Pham |
| Fondation pour la Recherche sur Alzheimer | PhD fellowship | Benjamin Le Gac |

The funders had no role in study design, data collection and interpretation, or the decision to submit the work for publication.

### Author contributions

Cuong Pham, Data curation, Formal analysis, Funding acquisition, Investigation, Methodology, Writing – original draft, Writing – review and editing; Yuji Komaki, Resources, Data curation, Formal analysis, Investigation, Visualization, Methodology, Writing – original draft, Writing – review and editing; Anna Deàs-Just, Data curation, Formal analysis, Investigation, Visualization, Methodology, Writing – original draft; Benjamin Le Gac, Data curation, Formal analysis, Funding acquisition, Investigation, Visualization, Methodology, Writing – original draft; Christine Mouffle, Agnès Chaperon, Resources, Methodology; Clara Franco, Data curation, Formal analysis, Investigation, Visualization, Methodology; Vincent Vialou, Resources, Data curation, Supervision, Methodology, Writing – original draft, Writing – review and editing; Tomokazu Tsurugizawa, Bruno Cauli, Dongdong Li, Conceptualization, Resources, Data curation, Software, Formal analysis, Supervision, Funding acquisition, Validation, Investigation, Visualization, Methodology, Writing – original draft, Project administration, Writing – review and editing

### Author ORCIDs

Yuji Komaki  https://orcid.org/0000-0002-0250-0354
Tomokazu Tsurugizawa  https://orcid.org/0000-0003-3194-578X
Bruno Cauli  https://orcid.org/0000-0003-1471-4621
Dongdong Li  https://orcid.org/0000-0002-6731-4771

## Ethics

All procedures using animals were carried out in strict accordance with French regulations (Rural Code R214/87 to R214/130) and conformed to the ethical recommendations of the European Economic Community (Directive 86/609/EEC) and the National Charter French on ethics in animal experimentation. All protocols were approved by the Charles Darwin ethics committee and submitted to the French Ministry of Education and Research (Approval 2015 061011367540 APAFIS#573-2015061011367547 v1).

Reviewer #1 (Public review): https://doi.org/10.7554/eLife.95873.3.sa1
Reviewer #3 (Public review): https://doi.org/10.7554/eLife.95873.3.sa2
Author response https://doi.org/10.7554/eLife.95873.3.sa3

## Additional files

### Supplementary files

• MDAR checklist

• Source code 1. Matlab scripts used for signal analysis. Calcium signal analysis by spatial-temporal correlation screening. The method principle has been described in *Pham et al., 2020*, which identified the active regions of interest (ROIs) of astrocytes displaying dynamic Ca$^{2+}$ signals.

### Data availability

All data generated or analysed during this study are included in the manuscript and supplementary files. Source code for calcium signal analysis is provided online.

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
