## [Editor Report · eLife Assessment]

In this work, the authors propose that astrocytic aquaporin 4 (AQP4) is the main pathway for tonic water efflux, without which astrocytes undergo cell swelling. These findings are **important**, because they shed light on key molecular mechanisms implicated with the regulation of brain water homeostasis. The authors use a broad set of experimental tools (e.g., acute brain slices, in vivo recording, and diffusion-weighted MRI) but the evidence remains **incomplete** without ruling out non-specific effects of TGN-020, and without evidence that changes in sulforhodamine B fluorescence can be used as reliable readouts of cell volume dynamics.

---

## [Referee Report · Reviewer #1 (Public review)]

Summary:

Pham and colleagues provide an illuminating investigation of aquaporin-4 water flux in the brain utilizing ex vivo and in vivo techniques. The authors first show in acute brain slices, and in vivo with fiber photometry, SRB loaded astrocytes swell after inhibition of AQP4 with TGN-020, indicative of tonic water efflux from astrocytes in physiological conditions. Excitingly, they find that TGN-020 increased the ADC in DW-MRI in a region-specific manner, potentially due to AQP4 density. The resolution of the DW-MRI cannot distinguish between intracellular or extracellular compartments, but the data point to an overall accumulation of water in the brain with AQP4 inhibition. These results provide further clarity on water movement through AQP4 in health and disease.

Overall, the data support the main conclusions of the article, with some room for more detailed treatment of the data to extend the findings.

Strengths:

The authors have a thorough investigation of AQP4 inhibition in acute brain slices. The demonstration of tonic water efflux through AQP4 at baseline is novel and important in and of itself. Their further testing of TGN-020 in hyper- and hypo-osmotic solutions shows the expected reduction of swelling/shrinking with AQP4 blockade.

Their experiment with cortical spreading depression further highlights the importance of water efflux from astrocytes via AQP4 and transient water fluxes as a result of osmotic gradients. Inhibition of AQP4 increases the speed of tissue swelling, pointing to a role in efflux of water from the brain.

The use of DW-MRI provides a non-invasive measure of water flux after TGN-020 treatment.

Weaknesses:

The authors specifically use GCaMP6 and light sheet microscopy to image their brain sections in order to identify astrocytic microdomains. However, their presentation of the data neglects a more detailed treatment of the calcium signaling. It would be quite interesting to see whether these calcium events are differentially affected by AQP4 inhibition based on their cellular localization (ie. processes vs. soma vs. vascular endfeet which all have different AQP4 expression).

The authors show the inhibition of AQP4 with TGN-020 shortens the onset time of the swelling associated with cortical spreading depression in brain slices. However, they do not show quantification for much of the other features of the CSD swelling, (ie. the duration of swelling, speed of swelling, recovery from swelling)

Comments on revised version:

The authors have addressed these suggestions as additional supplementary figures. Notably they find increased calcium signaling and stronger inhibition of calcium signaling by TGN-020 in astrocytic endfeet, where AQP4 is enriched.

Significance:

AQP4 is a bidirectional water channel that is constitutively open, thus water flux through it is always regulated by local osmotic gradients. Still, characterizing this water flux has been challenging, as the AQP4 channel is incredibly water selective. The authors here present important data showing that application of TGN-020 alone causes astrocytic swelling, indicating that there is constant efflux of water from astrocytes via AQP4 in basal conditions. This has been suggested before, as the authors rightfully highlight in their discussion, but the evidence had previously come from electron microscopy data from genetic knockout mice.

AQP4 expression has been linked with glymphatic circulation of cerebrospinal fluid through perivascular spaces since its rediscovery in 2012 [1]. Further studies of aging[2], genetic models[3], and physiological circadian variation[4], have revealed it is not simply AQP4 expression but AQP4 polarization to astrocytic vascular endfeet that is imperative for facilitating glymphatic flow. Still a lingering question in the field is how AQP4 facilitates fluid circulation. This study represents an important step in our understanding of AQP4's function, as basal efflux of water via AQP4 might promote clearance of interstitial fluid to allow influx of cerebrospinal fluid into the brain. Beyond glymphatic fluid circulation, clearly AQP4 dependent volume changes will differentially alter astrocytic calcium signaling and, in turn, neuronal activity.

(1) Iliff, J.J., et al., A Paravascular Pathway Facilitates CSF Flow Through the Brain Parenchyma and the Clearance of Interstitial Solutes, Including Amyloid β. Sci Transl Med, 2012. 4(147): p. 147ra111.

(2) Kress, B.T., et al., Impairment of paravascular clearance pathways in the aging brain. Ann Neurol, 2014. 76(6): p. 845-61.

(3) Mestre, H., et al., Aquaporin-4-dependent Glymphatic Solute Transport in the Rodent Brain. eLife, 2018. 7.

(4) Hablitz, L., et al., Circadian control of brain glymphatic and lymphatic fluid flow. Nature communications, 2020. 11(1).

---

## [Referee Report · Reviewer #3 (Public review)]

Summary:

In this manuscript, the Authors propose that astrocytic water channel AQP4 represents the dominant pathway for tonic water efflux without which astrocytes undergo cell swelling. The authors measure changes in astrocytic sulforhodamine B fluorescence as the proxy for cell volume dynamics. Using this approach, they have performed a technically elegant series of ex vivo and in vivo experiments exploring changes in astrocytic volume "signal" in response to the AQP4 inhibitor TGN-020 and/or neuronal stimulation. The key findings are that TGN-020 produces an apparent swelling of astrocytes and modifies astrocytic cell volume dynamics after spreading depolarizations. This study is perceived as potentially highly significant. However, several technical caveats could be considered better and perhaps addressed through additional experiments.

Strengths:

(1) This is a technically sound study, in which the Authors employed a number of complementary ex vivo and in vivo techniques. The presented results are of interest to the field and potentially highly significant.

(2) The innovative use of sulforhodamine B for in situ measurements of astrocyte cell volume dynamics is thoroughly validated in brain slices by quantifying changes in sulforhodamine fluorescence in response to hypoosmotic and hyperosmotic media.

(3) The combination of cell volume measurements with registering functional outcomes in both astrocytes and neurons (cell-specific GCaMP6 signaling) is appropriate and adds to the significance of the work.

(4) The use of ChR2 optogenetics for producing spreading depolarization allows to avoid many complications of chemical manipulations and much appreciated.

Remaining limitations:

(1) In the opinion of this reviewer, the effects of TGN-020 are not entirely consistent with the current knowledge on water permeability in astrocytes and the relative contribution of AQP4 to this process.

Specifically, genetic deletion of AQP4 reduces plasmalemmal water permeability in astrocytes by ~two-three-fold (when measured at 37oC, E. Solenov et al., AJP-Cell, 2004). This difference is significant but thought to have limited impact on steady-state water distribution. To the best of this reviewer's knowledge, cultured AQP4-null astrocytes do not show changes in degree of hypoosmotic swelling or hyperosmotic shrinkage. Thus, the findings of Solenov et al. are not (entirely) congruent with the conclusions of the current manuscript.

Also, as noted by the Authors, the AQP4 knockout does not modify astrocytes swelling induced by hypoosmotic solution in brain slices (T.R. Murphy et al., Front Neurosci., 2017), further suggesting that AQP4 is not a significant rate-limiting factor for water movement across astrocyte membranes.

The Authors do discuss the above-mentioned discrepancies and explain them by the context-dependent changes in water fluxes. Nevertheless, with these caveats in mind, it would be highly desirable to utilize an independent method measuring astrocytic volume and extracellular volume fraction.

(2) As noted by this reviewer and now discussed by the Authors, changes in ADC signal (presented in in Fig. 5) may be confounded by the previously reported TGN-020-induced hyphemia (e.g., H. Igarashi et al., NeuroReport, 2013) and/or changes water fluxes across pia matter which is highly enriched in AQP4. If this is the case, the proposed brain water accumulation may be explained by factors other than astrocytic water homeostasis. This caveat certainly deserves further experimental exploration.

---

## [Author Response]

The following is the authors’ response to the current reviews.

Many thanks to the editors for the reviewing of the revised manuscript.

We are very grateful to the Reviewers for their time and for the appreciation of the revision.

We thank the Reviewer 3 for acknowledging the use of sulforhodamine B (SRB) fluorescence as a real-time readout of astrocyte volume dynamics. Experimental data in brain slices were provided to validate this approach.

The incomplete matching of our observation with early reported data in cultured astrocytes (e.g., Solenov et al., AJP-Cell, 2004), might reflect certain of their properties differing from the slice/in vivo counterparts as discussed in the manuscript.

The study (T.R. Murphy et al., Front Cell Neurosci., 2017) showed that AQP4 knockout increased astrocyte swelling extent in response to hypoosmotic solution in brain slices (Fig 9), and discussed '... AQP4 can provide an efficient efflux pathway for water to leave astrocytes.’ Correspondingly, our data suggest that AQP4 mediate astrocyte water efflux in basal conditions.

We have discussed the study (Igarashi et al., NeuroReport 2013); our current data would help to understand the cellular mechanisms underlying the finding of Igarashi et al.

The following is the authors’ response to the original reviews.

**Public Reviews:**

**Reviewer #1 (Public Review):**
Summary:Pham and colleagues provide an illuminating investigation of aquaporin-4 water flux in the brain utilizing ex vivo and in vivo techniques. The authors first show in acute brain slices, and in vivo with fiber photometry, SRB-loaded astrocytes swell after inhibition of AQP4 with TGN-020, indicative of tonic water efflux from astrocytes in physiological conditions. Excitingly, they find that TGN-020 increases the ADC in DW-MRI in a region-specific manner, potentially due to AQP4 density. The resolution of the DW-MRI cannot distinguish between intracellular or extracellular compartments, but the data point to an overall accumulation of water in the brain with AQP4 inhibition. These results provide further clarity on water movement through AQP4 in health and disease.Overall, the data support the main conclusions of the article, with some room for more detailed treatment of the data to extend the findings.Strengths:The authors have a thorough investigation of AQP4 inhibition in acute brain slices. The demonstration of tonic water efflux through AQP4 at baseline is novel and important in and of itself. Their further testing of TGN-020 in hyper- and hypo-osmotic solutions shows the expected reduction of swelling/shrinking with AQP4 blockade.Their experiment with cortical spreading depression further highlights the importance of water efflux from astrocytes via AQP4 and transient water fluxes as a result of osmotic gradients. Inhibition of AQP4 increases the speed of tissue swelling, pointing to a role in the efflux of water from the brain.The use of DW-MRI provides a non-invasive measure of water flux after TGN-020 treatment.

We thank the reviewer for the insightful comments.

Weaknesses:The authors specifically use GCaMP6 and light sheet microscopy to image their brain sections in order to identify astrocytic microdomains. However, their presentation of the data neglects a more detailed treatment of the calcium signaling. It would be quite interesting to see whether these calcium events are differentially affected by AQP4 inhibition based on their cellular localization (ie. processes vs. soma vs. vascular end feet which all have different AQP4 expressions).

Following the suggestion, we provide new data on the effect of AQP4 inhibition on spontaneous calcium signals in perivascular astrocyte end-feet. As shown now in Fig.S2, acute application of TGN020 induced Ca2+ oscillations in astrocyte end-feet regions where the GCaMP6 labeling lines the profile of the blood vessel. It is noted that on average, the strength of basal Ca2+ signals in the end-feet is higher than that observed across global astrocyte territories (4.65 ± 0.55 vs. 1.45 ± 0.79, p < 0.01), as does the effect of TGN (8.4 ± 0.62 vs. 6.35 ± 0.97, p < 0.05; Fig S2 vs. Fig 2B). This likely reflects the enrichment of AQP4 in astrocyte end-feet. We describe the data in Fig.S2, and on page 8, line 20 – 23.

We now use the transgenic line GLAST-GCaMP6 for cytosolic GCaMP6 expression in astrocytes. Spontaneous calcium signals, reflected by transient fluorescence rises, occur in discrete micro-domains whereas the basal GCaMP6 fluorescence in the soma is weak. In the present condition, it is difficult to unambiguously discriminate astrocyte soma from the highly intermingled processes.

The authors show the inhibition of AQP4 with TGN-020 shortens the onset time of the swelling associated with cortical spreading depression in brain slices. However, they do not show quantification for many of the other features of CSD swelling, (ie. the duration of swelling, speed of swelling, recovery from swelling).

Regarding the features of the CSD swelling, we have performed new analysis to quantify the duration of swelling, speed of swelling and the recovery time from swelling in control condition and in the presence of TGN-020. The new analysis is now summarized in Fig. S5. Blocking AQP4 with TGN-020 increases the swelling speed, prolongs the duration of swelling and slows down the recovery from swelling, confirming our observation that acute inhibition of AQP4 water efflux facilitates astrocyte swelling while restrains shrinking. We describe the result on page 11, line 19-21.

Significance:AQP4 is a bidirectional water channel that is constitutively open, thus water flux through it is always regulated by local osmotic gradients. Still, characterizing this water flux has been challenging, as the AQP4 channel is incredibly water-selective. The authors here present important data showing that the application of TGN-020 alone causes astrocytic swelling, indicating that there is constant efflux of water from astrocytes via AQP4 in basal conditions. This has been suggested before, as the authors rightfully highlight in their discussion, but the evidence had previously come from electron microscopy data from genetic knockout mice.AQP4 expression has been linked with the glymphatic circulation of cerebrospinal fluid through perivascular spaces since its rediscovery in 2012 [1]. Further studies of aging[2], genetic models[3], and physiological circadian variation[4] have revealed it is not simply AQP4 expression but AQP4 polarization to astrocytic vascular endfeet that is imperative for facilitating glymphatic flow. Still, a lingering question in the field is how AQP4 facilitates fluid circulation. This study represents an important step in our understanding of AQP4's function, as the basal efflux of water via AQP4 might promote clearance of interstitial fluid to allow an influx of cerebrospinal fluid into the brain. Beyond glymphatic fluid circulation, clearly, AQP4-dependent volume changes will differentially alter astrocytic calcium signaling and, in turn, neuronal activity.(1) Iliff, J.J., et al., A Paravascular Pathway Facilitates CSF Flow Through the Brain Parenchyma and the Clearance of Interstitial Solutes, Including Amyloid β. Sci Transl Med, 2012. 4(147): p. 147ra111.(2) Kress, B.T., et al., Impairment of paravascular clearance pathways in the aging brain. Ann Neurol, 2014. 76(6): p. 845-61.(3) Mestre, H., et al., Aquaporin-4-dependent Glymphatic Solute Transport in the Rodent Brain. eLife, 2018. 7.(4) Hablitz, L., et al., Circadian control of brain glymphatic and lymphatic fluid flow. Nature Communications, 2020. 11(1).

We thank the reviewer in acknowledging the significance of our study and the functional implication in brain glymphatic system. We have now highlighted the mentioned studies as well as the potential implication glymphatic fluid circulation (page 4, line 9-10; page 5, line 1-3; and page 19, line 3-10).

**Reviewer #2 (Public Review):**
Summary:The paper investigates the role of astrocyte-specific aquaporin-4 (AQP4) water channel in mediating water transport within the mouse brain and the impact of the channel on astrocyte and neuron signaling. Throughout various experiments including epifluorescence and light sheet microscopy in mouse brain slices, and fiber photometry or diffusion-weighted MRI in vivo, the researchers observe that acute inhibition of AQP4 leads to intracellular water accumulation and swelling in astrocytes. This swelling alters astrocyte calcium signaling and affects neighboring neuron populations. Furthermore, the study demonstrates that AQP4 regulates astrocyte volume, influencing mainly the dynamics of water efflux in response to osmotic challenges or associated with cortical spreading depolarization. The findings suggest that AQP4-mediated water efflux plays a crucial role in maintaining brain homeostasis, and indicates the main role of AQP4 in this mechanism. However authors highlight that the report sheds light on the mechanisms by which astrocyte aquaporin contributes to the water environment in the brain parenchyma, the mechanism underlying these effects remains unclear and not investigated. The manuscript requires revision.Strengths:The paper elucidates the role of the astrocytic aquaporin-4 (AQP4) channel in brain water transport, its impact on water homeostasis, and signaling in the brain parenchyma. In its idea, the paper follows a set of complimentary experiments combining various ex vivo and in vivo techniques from microscopy to magnetic resonance imaging. The research is valuable, confirms previous findings, and provides novel insights into the effect of acute blockage of the AQP4 channel using TGN-020.

We thank the reviewer for the constructive comments.

Weaknesses:Despite the employed interdisciplinary approach, the quality of the manuscript provides doubts regarding the significance of the findings and hinders the novelty claimed by the authors. The paper lacks a comprehensive exploration or mention of the underlying molecular mechanisms driving the observed effects of astrocytic aquaporin-4 (AQP4) channel inhibition on brain water transport and brain signaling dynamics. The scientific background is not very well prepared in the introduction and discussion sections. The important or latest reports from the field are missing or incompletely cited and missconcluded. There are several citations to original works missing, which would clarify certain conclusions. This especially refers to the basis of the glymphatic system concept and recently published reports of similar content. The usage of TGN-020, instead of i.e. available AER-270(271) AQP4 blocker, is not explained. While employing various experimental techniques adds depth to the findings, some reasoning behind the employed techniques - especially regarding MRI - is not clear or seemingly inaccurate. Most of the time the number of subjects examined is lacking or mentioned only roughly within the figure captions, and there are lacking or wrongly applied statistical tests, that limit assessment and reproducibility of the results. In some cases, it seems that two different statistical tests were used for the same or linked type of data, so the results are contradictory even though appear as not likely - based on the figures. Addressing these limitations could strengthen the paper's impact and utility within the field of neuroscience, however, it also seems that supplementary experiments are required to improve the report.

The current data hint at a tonic water efflux from astrocyte AQP4 in physiological condition, which helps to understand brain water homeostasis and the functional implication for the glymphatic system. The underlying molecular and cellular mechanisms appear multifaceted and functionally interconnected, as discussed (page 14 line 8 –page 15, line 3). We agree that a comprehensive exploration will further advance our understanding.

The introduction and discussion are now strengthened by incorporating the important advances in glymphatic system while highlighting the relevant studies.

The use of TGN-020 was based on its validation by wide range of ex vivo and in vivo studies including the use of heterologous expression system and the AQP4 KO mice. The validation of AER-270(271, the water soluble prodrug) using AQP4 KO mice is reported recently (Giannetto et al., 2024). AER-271 was noted to impact brain water ADC (apparent diffusion coefficient evaluated by diffusion-weighted MRI) in AQP4 KO mice ~75 min after the drug application (Giannetto et al., 2024). This likely reflects that AER270(271) is also an inhibitor for κΒ nuclear factor (NF-κΒ) whose inhibition could reduce CNS water content independent of AQP4 targeting (Salman et al., 2022). In addition, the inhibition efficiency of AER-270(271) seems lower than TGN-020 (Farr et al., 2019; Giannetto et al., 2024; Huber et al., 2009; Salman et al., 2022). We have now supplemented this information in the manuscript (page 7, line 1-6 and page15, line 7-17).

The description on the DW-MRI is now updated (page 4, line 10-14).

We also performed new experiments and data analysis as described in a point-to-point manner below in the section ‘*Recommendations For The Authors*’.

**Reviewer #3 (Public Review):**
Summary:In this manuscript, the authors propose that astrocytic water channel AQP4 represents the dominant pathway for tonic water efflux without which astrocytes undergo cell swelling. The authors measure changes in astrocytic sulforhodamine fluorescence as the proxy for cell volume dynamics. Using this approach, they perform a technically elegant series of ex vivo and in vivo experiments exploring changes in astrocytic volume in response to AQP4 inhibitor TGN-020 and/or neuronal stimulation. The key finding is that TGN-020 produces an apparent swelling of astrocytes and modifies astrocytic cell volume regulation after spreading depolarizations. Additionally, systemic application of TGN-020 produced changes in diffusion-weighted MRI signal, which the authors interpret as cellular swelling. This study is perceived as potentially significant. However, several technical caveats should be strongly considered and perhaps addressed through additional experiments.Strengths:(1) This is a technically elegant study, in which the authors employed a number of complementary ex vivo and in vivo techniques to explore functional outcomes of aquaporin inhibition. The presented data are potentially highly significant (but see below for caveats and questions related to data interpretation).(2) The authors go beyond measuring cell volume homeostasis and probe for the functional significance of AQP4 inhibition by monitoring Ca2+ signaling in neurons and astrocytes (GCaMP6 assay).(3) Spreading depolarizations represent a physiologically relevant model of cellular swelling. The authors use ChR2 optogenetics to trigger spreading depolarizations. This is a highly appropriate and much-appreciated approach.

We thank the reviewer for the effort in evaluating our work.

Weaknesses:(1) The main weakness of this study is that all major conclusions are based on the use of one pharmacological compound. In the opinion of this reviewer, the effects of TGN-020 are not consistent with the current knowledge on water permeability in astrocytes and the relative contribution of AQP4 to this process.Specifically: Genetic deletion of AQP4 in astrocytes reduces plasmalemmal water permeability by ~two-three-fold (when measured a 37oC, Solenov et al., AJP-Cell, 2004). This is a significant difference, but it is thought to have limited/no impact on water distribution. Astrocytic volume and the degree of anisosmotic swelling/shrinkage are unchanged because the water permeability of the AQP4null astrocytes remains high. This has been discussed at length in many publications (e.g., MacAulay et al., Neuroscience, 2004; MacAulay, Nat Rev Neurosci, 2021) and is acknowledged by Solenov and Verkman (2004).Keeping this limitation in mind, it is important to validate astrocytic cell volume changes using an independent method of cell volume reconstruction (diameter of sulforhodamine-labeled cell bodies? 3D reconstruction of EGFP-tagged cells? Else?)

Solenov and coll. used the calcein quenching assay and KO mice demonstrating AQP4 as a functional water channel in cultured astrocytes (Solenov et al., 2004). AQP4 deletion reduced both astrocyte water permeability and the absolute amplitude of swelling over comparable time, and also slowed down cell shrinking, which overall parallels our results from acute AQP4 blocking. Yet in Solenovr’s study, the time to swelling plateau was prolonged in AQP4 KO astrocytes, differing from our data from the pharmacological acute blocking. This discrepancy may be due to compensatory mechanisms in chronic AQP4 KO, or reflect the different volume responses in cultured astrocytes from brain slices or in vivo results as suggested previously (Risher et al., 2009).

Soma diameter might be an indicator of cell volume change, yet it is challenging with our current fluorescence imaging method that is diffraction-limited and insufficient to clearly resolve the border of the soma in situ. In addition, the lateral diameter of cell bodies may not faithfully reflect the volume changes that can occur in all three dimensions. Rapid 3D imaging of astrocyte volume dynamics with sufficient high Z-axis resolution appears difficult with our present tools.

We have now accordingly updated the discussion with relevant literatures being cited (page 17 line 14 – page 18, line 3).

(2) TGN-020 produces many effects on the brain, with some but not all of the observed phenomena sensitive to the genetic deletion of AQP4. In the context of this work, it is important to note that TGN020 does not completely inhibit AQP4 (70% maximal inhibition in the original oocyte study by Huber et al., Bioorg Med Chem, 2009). Thus, besides not knowing TGN-020 levels inside the brain, even"maximal" AQP4 inhibition would not be expected to dramatically affect water permeability in astrocytes.This caveat may be addressed through experiments using local delivery of structurally unrelated AQP4 blockers, or, preferably, AQP4 KO mice.

It is an important point that TGN-020 partially blocks AQP4, implying the actual functional impact of AQP4 per se might be stronger than what we observed. TGN provides a means to acutely probe AQP4 function in situ, still we agree, its limitation needs be acknowledged. We mention this now on page 15, line 7-9 and 14-17.

We agree that local delivery of an alternative blocker will provide additional information. Meanwhile, local delivery requires the stereotaxic implantation of cannula, which would cause inflammations to surrounding astrocytes (and neurons). The recently introduced AQP4 blocker AER-270(271) has received attention that it influences brain water dynamics (ADC in DW-MRI) in AQP4 KO mice (Giannetto et al., 2024), recalling that AER-270(271) is also an inhibitor for κΒ nuclear factor (NF-κΒ). This pathway can potentially perturb CNS water content and influence brain fluid circulation, in an AQP4independent manner (Salman et al., 2022). The inhibition efficiency on mouse AQP4 of AER-270 (~20%, Farr et al., 2019; Salman et al., 2022) appears lower than TGN-020 (~70%, Huber et al., 2009).

We chose to use the pharmacological compound to achieve acute blocking of AQP4 thereby avoiding the chronic genetics-caused alterations in brain structural, functional and water homeostasis. Multiple lines of evidence including the recent study (Gomolka et al., 2023), have shown that AQP4 KO mice alters brain water content, extracellular space and cellular structures, which raises concerns to use the transgenic mouse to pinpoint the physiological functions of the AQP4 water channel.

We have now mentioned the concerns on AQP4 pharmacology by supplementing additional literatures in the field (page 15, line 8-18).

(3) This reviewer thinks that the ADC signal changes in Figure 5 may be unrelated to cellular swelling. Instead, they may be a result of the previously reported TGN-020-induced hyphemia (e.g., H. Igarashi et al., NeuroReport, 2013) and/or changes in water fluxes across pia matter which is highly enriched in AQP4. To amplify this concern, AQP4 KO brains have increased water mobility due to enlarged interstitial spaces, rather than swollen astrocytes (RS Gomolka, eLife, 2023). Overall, the caveats of interpreting DW-MRI signal deserve strong consideration.

The previous observation show that TGN-020 increases regional cerebral blood flow in wild-type mice but not in AQP4 KO mice (Igarashi et al., 2013). Our current data provide a possible mechanism explanation that TGN-020 blocking of astrocyte AQP4 causes calcium rises that may lead to vasodilation as suggested previously (Cauli and Hamel, 2018). We now add updates to the discussion on page 15, line 3-7.

We are in line with the reviewer regarding the structural deviations observed with the AQP4 KO mice

(Gomolka et al., 2023), now mentioned on page 19, line 3-5. Following the Reviewer’s suggestion, we have also updated the interpretation of the DW-MRI signal and point that in addition to being related to the astrocyte swelling, the ADC signal changes may also be caused by indirect mechanisms, such as the transient upregulation of other water-permeable pathways in compensating AQP4 blocking. We now describe this alternative interpretation and the caveats of the DW-MRI signals (page 20, line 1-8).

**Recommendations for the authors:**

**Reviewer #1 (Recommendations For The Authors):**
Private recommendationsMy more broad experimental suggestions are in the "weaknesses" section. Some minor points that would improve the manuscript are included below:(1) A more detailed explanation for why SRB fluorescence reflects the astrocyte volume changes, whereas typical intracellular GFP does not.

As an engineered fluorescence protein, the GFP has been used to tag specific type of cells. Meanwhile, as a relatively big protein (MW, 26.9 kDa), the diffusion rate of EGFP is expected to be much less than SRB, a small chemical dye (MW, 558.7 Da). Also, the IP injection of SRB enables geneticsless labeling of brain astrocytes, so to avoid the influence of protein overexpression on astrocyte volume and water transport responses. We have now stated this point in the manuscript (page 13, line 21 – page 14, line 4).

(2) Figure 1 panel B should have clear labels on the figure and a description in the legend to delineate which part of the panel refers to hyper- or hypo-osmotic treatment.

We have now updated the figure and the legend.

(3) For Figure 2, what is the rationale for analyzing the calcium signaling data between the cell types differently?

We analyzed calcium micro-domains for astrocytes as their spontaneous signals occur mainly in discrete micro-domains (Shigetomi et al., 2013). While for neurons, we performed global analysis by calculating the mean fluorescence of imaging field of view, because calcium signal changes were only observed at global level rather than in micro-domains. This information is now included (page 24, line1820).

(4) For Figure 3, the authors mention that TGN-020 likely caused swelling prior to the hypotonic solution administration. Do they have any measurements from these experiments prior to the TGN-020 application to use as a "true baseline" volume?

The current method detects the relative changes in astrocyte volume (i.e., transmembrane water transport), which nevertheless is blind to the absolute volume value. We have no readout on baseline volumes.

(5) For Figures 3 and 4, did the authors see any evidence for regulatory volume decrease? And is this impaired by TGN-020? It is a well-characterized phenomenon that astrocytes will open mechanosensitive channels to extrude ions during hypo-osmotic induced swelling. This process is dependent on AQP4 and calcium signaling [5]

Mola and coll. provided important results demonstrating the role of AQP4 in astrocyte volume regulation (Mola et al., 2016). In the present study in acute brain slices, when we applied hypotonic solution to induce astrocyte swelling, our protocol did not reveal rapid regulatory volume decrease (e.g., Fig. 3D). When we followed the volume changes of SRB-labeled astrocytes during optogenetically induced CSD, we observed the phase of volume decrease following the transient swelling (Fig. 4F), where the peak amplitude and the degree of recovery were both reduced by inhibiting AQP4 with TGN020. These data imply that regulatory astrocyte volume decrease may occur in specific conditions, which intriguingly has been suggested to be absent in brain slices and in vivo (e.g., Risher et al., 2009). We have not specifically investigated this phenomenon, and now briefly discuss this point on page18 line 6-14.

(6) Figure 5 box plots do not show all data points, could the authors modify to make these plots show all the animals, or edit the legend to clarify what is plotted?

We have now updated the plot and the legend. This plot is from all animals (n = 7 per condition).

(7) pg. 9 line 6, there is a sentence that seems incomplete or otherwise unfinished. "We first followed the evoked water efflux and shrinking induced by hypertonic solution while."

Fixed (now, page 9 line 17-18).

(8) During the discussion on pg 13 line 11, it may be more clear to describe this as the cotransport of water into the cells with ions/metabolites as reviewed by Macaulay 2021 [6].

We agree; the text is modified following this suggestion (now page14, line 12-13).

(1) Iliff, J.J., et al., A Paravascular Pathway Facilitates CSF Flow Through the Brain Parenchyma and the Clearance of Interstitial Solutes, Including Amyloid β. Sci Transl Med, 2012. 4(147): p. 147ra111.(2) Kress, B.T., et al., Impairment of paravascular clearance pathways in the aging brain. Ann Neurol, 2014. 76(6): p. 845-61.(3) Mestre, H., et al., Aquaporin-4-dependent Glymphatic Solute Transport in the Rodent Brain. eLife, 2018. 7.(4) Hablitz, L., et al., Circadian control of brain glymphatic and lymphatic fluid flow. Nature Communications, 2020. 11(1).(5) Mola, M., et al., The speed of swelling kinetics modulates cell volume regulation and calcium signaling in astrocytes: A different point of view on the role of aquaporins. Glia, 2016. 64(1).(6) MacAulay, N., Molecular mechanisms of brain water transport. Nat Rev Neurosci, 2021. 22(6): p. 326-344.

We thank the reviewer. These important literatures are now supplemented to the manuscript together with the corresponding revisions.

**Reviewer #2 (Recommendations For The Authors):**
In its concept, the paper is interesting and provides additional value - however, it requires revision.Below, I provide the following remarks for the following sections/ pages/lines:ABSTRACT/page 2 (remarks here refer to the rest of the manuscript, where these sentences are repeated):- It seems that the 'homeostasis' provides not only physical protection, but also determines the diffusion of chemical molecules...' Please correct the sentence as it is grammatically incorrect.

It is now corrected (page 2, line 1).

- The term 'tonic water' is not clear. I understand, after reading the paper, that it is about tonicity of the solutes injected into the mouse.

We use the term ‘tonic’ to indicate that in basal conditions, a constant water efflux occurs through the APQ4 channel.

- 'tonic aquaporin water efflux maintains volume equilibrium' - I believe it is about maintaining volume and osmotic equilibrium?

This description is now refined (now page 2, line 10).

- It is not clear whether the tonic water outflow refers to the cellular level or outflow from the brain parenchyma (i.e., glymphatic efflux)

It refers to the cellular level.

INTRODUCTION/page 3:- 'clearance of waste molecules from the brain as described in the glymphatic system' - The original papers describing the phenomena are not cited: Iliff et al. 2012, 2013, Mestre et al. 2018, as well as reviews by Nedergaard et al.

Indeed. We have now cited these key literatures (now page 4, line 10).

- 'brain water diffusion is the basis for diffusion-weighted magnetic resonance imaging (DW-MRI)' - The statement is wrong. it is the mobility of the water protons that DWI is based on, but not the diffusion of molecules in the brain. This should be clarified and based on the DW-MRI principle and the original works by Le Bihan from 1986, 1988, or 2015.

This sentence is now updated (page 4, line10-14).

- Similarly, I suggest correcting or removing the citations and the sentence part regarding the clinical use of DWI, as it has no value here. Instead, it would be worth mentioning what actually ADC reflects as a computational score, and what were the results from previous studies assessing glymphatic systems using DWI. This is especially important when considering the mislocalization of the AQP4 channel.

We now states recent studies using DW-MRI to evaluate glymphatic systems (page 4, line16-17).

- 'In the brain, AQP4 is predominantly expressed in astrocytes'-please review the citations. I suggest reading the work by Nielsen 1997, Nagelhus 2013, Wolburg 2011, and Li and Wang from 2017. To my best knowledge, in the brain AQP4 is exclusively expressed in astrocytes.

Thanks for the reviewer. It is described that while enriched in astrocytes, AQP4 is also expressed in ependymal cells lining the ventricles (e.g., (Mayo et al., 2023; Verkman et al., 2006)). ‘predominantly’ is now removed (page 4, line 21).

- The conclusion: ' Our finding suggests that aquaporin acts as a water export route in astrocytes in physiological conditions, so as to counterbalance the constitutive intracellular water accumulation caused by constant transmitter and ion uptake, as well as the cytoplasmic metabolism processes. This mechanism hence plays a necessary role in maintaining water equilibrium in astrocytes, thereby brain water homeostasis' seems to be slightly beyond the actual findings in the paper. I suggest clarifying according to the described phenomena.

We have now refined the conclusion sticking to the experimental observations (page 5, line16-18).

- The introduction lacks important information on existing AQP4 blockers and their effects, pros and cons on why to use TGN-020. Among others, I would refer to recent work by Giannetto et al 2024, as well as previous work of Mestre et al. 2018 and Gomolka et al. 2023.

We initiated the study by using TGN-020 as an AQP4 blocker because it has been validated by wide range of ex vivo and in vivo studies as documented in the text (page 7, line 1-6). We also update discussions on the recent advances in validating the AQP4 blocker AER-270(271) while citing the relevant studies (page 15, line 7-17).

RESULTS:- Page 5, lines 19-20: '...transport, we performed fluorescence intensity translated (FIT) imaging.' - this term was never introduced in the methods so it is difficult for the reader to understand it at first sight. -'To this end,' - it is not clear which action refers to 'this'. is it about previous works or the moment that the brain samples were ready for imaging? Please clarify, as it is only starting to be clear after fully reading the methods.

We now refine the description give the principle of our imaging method first, then explain the technical steps. To avoid ambiguity, the term ‘To this end’ is removed. The updated text is now on page 6, line 1-3.

- From page 6 onwards - all references to Figures lack information to which part of the figure subpanel the information refers (top/middle bottom or left/middle/right).

We apologize. The complementary indication is now added for figure citations when applicable.

- 'whereas water export and astrocyte shrinking upon hyperosmotic manipulation increased astrocyte fluorescence (Figure 1B). Hence, FIT imaging enables real-time recording of astrocyte transmembrane water transport and volume dynamics.' - this part seems to be undescribed or not clear in the methods.

We have now refined this description (page 6, line 19-20).

- Page 6, lines 17-22: TGN-020. In addition to the above, I suggest familiarizing also with the following works by Igarashi 2011. doi: 10.1007/s10072-010-0431-1, and by Sun 2022. doi: 10.3389/fimmu.2022.870029.

These studies are now cited (page 7, line 3-4).

- Page 7: ' AQP4 is a bidirectional channel facilitating... ' - AQP4 water channel is known as the path of least resistance for water transfer, please see Manley, Nature Medicine, 2000 and Papadopoulos, Faseb J, 2004.

This sentence is now updated (page 7, line 12-13).

- ' astrocyte AQP4 by TGN-020 caused a gradual decrease in SRB fluorescence intensity, indicating an intracellular water accumulation' - tissue slice experiment is a very valuable method. However it seems right, the experiment does not comment on the cell swelling that may occur just due to or as a superposition of tissue deterioration and the effect of TGN-020. The AQP4 channel is blocked, and the influx of water into astrocytes should be also blocked. Thus, can swelling be also a part of another mechanism, as it was also observed in the control group? I suggest this should be addressed thoroughly.

We performed this experiment in acute brain slices to well control the pharmacological environment and gain spatial-temporal information. Post slicing, the brain slices recovered > 1hr prior to recording, so that the slices were in a stable state before TGN-020 application as evidenced by the stable baseline. The constant decrease in the control trace is due to photobleaching which did not change its curve tendency in response to vehicle. TGN-020, in contrast, caused a down-ward change suggesting intracellular water accumulation and swelling.

The experiment was performed at basal condition without active water influx; a decrease in SRB fluorescence hints astrocyteintracellular water buildup. This result shows that in basal condition, astrocyte aquaporin mediates a constant (i.e., tonic) water efflux; its blocking causes intracellular water accumulation and swelling.

We have accordingly updated the description of this part (page 7, line 15-20).

- From the Figure 1 legend: Only 4 mice were subjected to the experiment, and only 1 mouse as a control. I suggest expanding the experiment and performing statistics including two-way ANOVA for data in panels B, C, and D, as no results of statistical tests confirm the significance of the findings provided.

The panel B confirms that cytosolic SRB fluorescence displays increasing tendency upon water efflux and volume shrinking, and vice versa. As for the panel C, the number of mice is now indicated. Also, the downward change in the SRB fluorescence was now respectively calculated for the phases prior and post to TGN (and vehicle) application, and this panel is accordingly updated. TGN-020 induced a declining in astrocyte SRB fluorescence, which is validated by t-test performed in MATLAB. To clarify, we now add cross-link lines to indicate statistical significance between the corresponding groups (Fig 1C, middle). As for panel D, we calculated the SRB fluorescence change (decrease) relative to the photobleaching tendency illustrated by the dotted line. The significance was also validated by t-test performed in MATLAB.

- Figure 1: Please correct the figure - pictures in panel A are low quality and do not support the specificity of SRB for astrocytes. Panels B-D are easier to understand if plotted as normal X/Y charts with associated statistical findings. Some drawings are cut or not aligned.

In GFAP-EGFP transgenic, astrocytes are labeled by EGFP. SRB labeling (red fluorescence) shows colocalization with EGFP-positive astrocytes, meanwhile not all EGFP-positive astrocytes are labeled by SRB. The PDF conversion procedure during the submission may also somehow have compromised image quality. We have tried to update and align the figure panels.

- Page 12: ' TGN-020 increased basal water diffusion within multiple regions including the cortex,

hippocampus and the striatum in a heterogeneous manner (Figure 5C).'

This sentence is updated now (page 12, line 12 – page13, line 2). It reads ‘The representative images reveal the enough image quality to calculate the ADC, which allow us to examine the effect of TGN-020 on water diffusion rate in multiple regions (Fig. 5C).’

- The expression of AQP4 within the brain parenchyma is known to be heterogenous. Please familiarize yourself with works by Hubbard 2015, Mestre 2018, and Gomolka 2023. A correlation between ADC score and AQP4 expression ROI-wise would be useful, but it is not substantial to conduct this experiment.

We thank the reviewer. This point is stressed on page 19, line 12-14.

DISCUSSION:- Most of the issues are commented on above, so I suggest following the changes applied earlier. -Page 16: 'We show by DW-MRI that water transport by astrocyte aquaporin is critical for brain water homeostasis.' This statement is not clear and does not refer to the actual impact of the findings. DWI is allowed only to verify the changes of ADC fter the application of TGN-020. I suggest commenting on the recent report by Giannetto 2024 here.

This sentence is now refined (page 19, line 1-2), followed by the updates commenting on the recent studies employing DW-MRI to evaluate brain fluid transport, including the work of (Giannetto et al., 2024) (page 19, line 3-10).

METHODS:- Page 18: no total number of mice included in all experiments is provided, as well as no clearly stated number of mice used in each experiment. Please correct.

We have now double checked the number of the mice for the data presented and updated the figure legends accordingly (e.g., updates in legends fig1, fig5, etc).

- Page 18, line 7: 'Axscience' is not a producer of Isoflurane, but a company offering help with scientific manuscript writing. If this company's help was used, it should be stated in the acknowledgments section. Reference to ISOVET should be moved from line 15 to line 7.

We apologize. We did not use external writing help, and now have removed the ‘Axcience’. The Isoflurane was under the mark ‘ISOVET’ from ‘Piramal’. This info is now moved up (page 21, line 11).

- Page 18, line 9: ' modified artificial cerebrospinal fluid (aCSF)'. Additional information on the reason for the modified aCSF would be useful for the reader.

In this modified solution, the concentration of depolarizing ions (Na+, Ca2+) was reduced to lower the potential excitotoxicity during the tissue dissection (i.e., injury to the brain) for preparing the brain slices. Extra sucrose was added to balance the solution osmolarity. This solution has been used previously for the dissection and the slicing steps in adult mice (Jiang et al., 2016). We now add this justification in the text and quote the relevant reference (page 21, line14-16).

- Page 19, line 6: a reasoning for using Tamoxifen would be helpful for the reader.

The Glast-CreERT2 is an inducible conditional mouse line that expresses Cre recombinase selectively in astrocytes upon tamoxifen injection. We now add this information in the text (page 22, line 10-11).

- Line 8 - 'Sigma'

Fixed.

- Line 7/8: It is not clear if ethanol is of 10% solution or if proportions of ethanol+tamoxifen to oil were of 1:9. The reasoning for each performed step is missing.

We have now clarified the procedure (page 22, line 11-15).

- Line 10: '/' means 'or'?

Here, we mean the bigenic mice resulting from the crossing of the heterozygous Cre-dependent GCaMP6f and Glast-CreERT2 mouse lines. We now modify it to ‘Glast-CreERT2::Ai95GCaMP6f//WT’, in consistence with the presentation of other mouse lines in our manuscript (page 22, line 16).

- Lines 22-23: being in-line with legislation was already stated at the beginning of the Methods so I suggest combining for clearance.

Done.

- Page 21, line 4: it is good to mention which printer was used, but it would be worth mentioning the material the chamber was printed from - was it ABS?

Yes. We add this info in the text now (page 24, line 5).

- Line 9 -'PI' requires spelling out.

It is ‘Physik Instrumente’, now added (page 24, line 10).

- Line 11-12: What is the reason for background subtraction - clearer delineation of astrocytes/ increasing SNR in post-processing, or because SRB signal was also visible and changing in the background over time? Was the background removed in each frame independently (how many frames)? How long was the time-lapse and was the F0 frame considered as the first frame acquired? The background signal should be also measured and plotted alongside the astrocytic signal, as a reference (Figure 1). This should be clarified so that steps are to be followed easily.

We sought to follow the temporal changes in SRB fluorescence signal. The acquired fluorescent images contain not only the SRB signals, but also the background signals consisting of for instance the biological tissue autofluorescence, digital camera background noise and the leak light sources from the environments. The value of the background signal was estimated by the mean fluorescence of peripheral cell-free subregions (15 × 15 µm²) and removed from all frames of time-lapse image stack. The traces shown in the figures reflect the full lengths of the time-lapse recordings. F0 was identified as the mean value of the 10 data points immediately preceding the detected fluorescence changes. The text is now updated (page 24 line 21 - page 25 line 5).

- Line 15: Was astrocyte image delineation performed manually or automatically? Where was the center of the region considered in the reference to the astrocyte image? It would be good to see the regions delineated for reference.

Astrocytes labeled by SRB were delineated manually with the soma taken as the center of the region of interest. We now exemplify the delineated region in Fig 1A, bottom.

- Page 22, line 2: 'x4 objective'.

Added (now, page 25, line 16).

- Line 3: 'barrels' - reference to publication or the explanation missing.

The relevant reference is now added on barrel cortex (Erzurumlu and Gaspar, 2020) (page 25, line 19-20).

- Line 19: were the coordinates referred to = bregma?

Yes. This info is now added (page 26, line 12).

- Line 20: was the habituation performed directly at the acquisition date? It is rather difficult to say that it was a habituation, but rather acute imaging. I suggest correcting, that mice were allowed to familiarize themselves with the setup for 30 minutes prior to the imaging start.In this context, although it is a very nice idea and experiment, the influence of acute stress in animals familiar with the setup only from the day of acquisition is difficult to avoid. It is a major concern, especially when considering norepinephrine as a master driver of neuronal and vascular activity through the brain, and strong activation of the hypothalamic-adrenal axis in response to acute stress. It is well known, that the response of monoamines is reduced in animals subjected to chronic v.s acute stress, but still larger than that if the stressor is absent.Major remark: The animals should, preferably, be imaged at least after 3 days of habituation based on existing knowledge. I suggest exploring the topic of the importance of habituation. It is difficult though, to objectively review these findings without considering stress and associated changes in vascular dynamics.

Many thanks for the reviewer to help to precise this information. The text is accordingly updated to describe the experiment (now page 26, line 14).

- Page 23, line 17: number of animals included in experiments missing.

The number of animals is added in Methods (page 27, line 12) and indicated in the legend of Figure 5.

- Line 18/19: were the respiratory effects observed after injection of saline or TGN-020? Since DWI was performed, the exclusion of perfusive flow on ADC is impossible.I suggest an additional experiment in n=3 animals per group, verifying the HR (and if possible BP) response after injection of TGN-020 and saline in mice.

The respiratory rate has been recorded. We added the averaged respiratory rate before and after injection of TGN-020 or saline (now, Fig. S6; page 13, line 5-6).

- Line 22: Please, provide the model of the scanner, the model of the cryoprobe, as well as the model of the gradient coil used, otherwise it is difficult to assess or repeat these experiments.

We have now added the information of MRI system in Methods section (page 27, line17-21).

- Page 24: line 3/4: although the achieved spatial resolution of DWI was good and slightly lower than desired and achievable due to limitations of the method itself as well as cryoprobe, it is acceptable for EPI in mice.Still, there is no direct explanation provided on the reasoning for using surface instead of volumetric coil, as well as on assuming an anisotropic environment (6 diffusion directions) for DWI measurements. This is especially doubtful if such a long echo-time was used alongside lower-thanpossible spatial resolution. Longer echo time would lower the SNR of the depicted signal but also would favor the depiction of signal from slow-moving protons and larger water pools. On the other hand, only 3 b-values were used, which is the minimum for ADC measurements, while a good research protocol could encompass at least 5 to increase the accuracy of ADC estimation and avoid undersampling between 250 and 1800 b-values. What was the reason for choosing this particular set of b-values and not 50, 600, and 2000? Besides, gradient duration time was optimally chosen, however, I have concerns about the decision for such a long gradient separation times.If the protocol could have been better optimized, the assessment could have been also performed in respiratory-gated mode, allowing minimization of the effects of one of the glymphatic system driving forces.Thus, I suggest commenting on these issues.

We chose the cryoprobe to increase the signal-to-noise ratio (SNR) in DW-MRI with long echo-time and high b-value. The volume coil has a more homogeneous SNR in the whole brain rather than the cryoprobe, but SNR should be reduced compared with cryoprobe. We confirmed that, even at the ventral part of the brain, the image quality of DW-MRI images was enough to investigate the ADC with cryoprobe (Fig. 5B-C). This is mentioned now in Methods (page 27, line 17-21).

We performed DW-MRI scanning for 5 min at each time-point using the condition of anisotropic resolution and 3 b-values, to investigate the time-course of ADC change following the injection of TGN020. Because the effect of TGN-020 appears about dozen of minutes post the injection (Igarashi et al., 2011), fast DW-MRI scanning is required. If isotropic DW-MRI with lower echo-time and more direction is used, longer scan time at each time point is required, maybe more than 1h. We agree that three bvalues is minimum to calculate the ADC and more b-values help to increase the accuracy. However, to achieve the temporal resolution so as to better catch the change of water diffusion, we have decided to use the minimum b-values. The previous study also validates the enough accuracy of DW-MRI with three b-values (Ashoor et al., 2019). Furthermore, previous study that used long diffusion time (> 20 ms) and long echo time (40 ms) shows the good mean diffusivity (Aggarwal et al., 2020), supporting that our protocol is enough to investigate the ADC. We have now updated the description (page 28 line 5-9). The reason why we choose the b = 250 and 1800 s/mm² is that 2000 s/mm² seems too high to get the good quality of image. In the previous study, we have optimized that ADC is measurable with b = 0, 250, and 1800 s/mm² (Debacker et al., 2020).

- Page 24, line 7: What was the post-processing applied for images acquired over 70 minutes? Did it consider motion-correction, co-registration, or drift-correction crucial to avoid pitfalls and mismatches in concluding data?

The motion correction and co-registration were explained in Methods (page 28, line 12-14).

Also, were these trace-weighted images or magnitude images acquired since DTI software was used for processing - while ADC fitting could be reliably done in Matlab, Python, or other software. Thus, was DSI software considering all 3 b-values or just used 0 and 1800 for the calculation of mean diffusivity for tractography (as ADC). The details should be explained.

DSIstudio was used with all three b values (b = 0, 250, and 1800 s/mm²) to calculate the ADC. We added the description in Methods (page 28, line 16-18).

To make sure that the results are not affected by the MR hardware, I suggest performing 3 control measurements in a standard water phantom, and presenting the results alongside the main findings.

Thanks for this suggestion. We have performed new experiments and now added the control measurement with three phantoms, that is water, undecane, and dodecane. These new data are summarized now in Fig. S7, showing the stability of ADC throughout the 70 min scanning. We have updated the description on Method part (page 28, line 9-11) and on the Results (page 13, line 6-8).

- Line 13: were the ROI defined manually or just depicted from previously co-registered Allen Brain atlas?

The ROIs of the cortex, the hippocampus, and the striatum were depicted with reference to Allen mouse brain atlas (https://scalablebrainatlas.incf.org/mouse/ABA12). This is explained in Methods (page 28, line 14-16).

- Line 10: why the average from 1st and 2nd ADC was not considered, since it would reduce the influence of noise on the estimation of baseline ADC?

We are sorry that it was a typo. The baseline was the average between 1st and 2nd ADC. We corrected the description (page 28, line 20).

STATISTIC:Which type of t-test - paired/unpaired/two samples was used and why? Mann-Whitney U-tets are used as a substitution for parametric t-tests when the data are either non-parametric or assuming normal distribution is not possible. In which case Bonferroni's-Holm correction was used? - I couldn't find any mention of any multiple-group analysis followed by multiple comparisons. Each section of the manuscript should have a description of how the quantitative data were treated and in which aim. I suggest carefully correcting all figures accordingly, and following the remarks given to the Figure 1.

We used unpaired t-test for data obtained from samples of different conditions. Indeed, MannWhitney U-test is used when the data are non-parametric deviating from normal distributions. Bonferroni-Holm correction was used for multiple comparisons (e.g., Fig. 4D-E).

**Reviewer #3 (Recommendations For The Authors):**
I think that the following statement is insufficient: "The authors commit to share data, documentation, and code used in analysis". My understanding is eLife expects that all key data to be provided in a supplement.

We thank the reviewer; we follow the publication guidelines of eLife.

References

Aggarwal, M., Smith, M.D., and Calabresi, P.A. (2020). Diffusion-time dependence of diffusional kurtosis in the mouse brain. Magn Reson Med 84, 1564-1578.

Ashoor, M., Khorshidi, A., and Sarkhosh, L. (2019). Estimation of microvascular capillary physical parameters using MRI assuming a pseudo liquid drop as model of fluid exchange on the cellular level. Rep Pract Oncol Radiother 24, 3-11.

Cauli, B., and Hamel, E. (2018). Brain Perfusion and Astrocytes. Trends in neurosciences 41, 409-413.

Debacker, C., Djemai, B., Ciobanu, L., Tsurugizawa, T., and Le Bihan, D. (2020). Diffusion MRI reveals in vivo and non-invasively changes in astrocyte function induced by an aquaporin-4 inhibitor. PLoS One 15, e0229702.

Erzurumlu, R.S., and Gaspar, P. (2020). How the Barrel Cortex Became a Working Model for Developmental Plasticity: A Historical Perspective. J Neurosci 40, 6460-6473.

Farr, G.W., Hall, C.H., Farr, S.M., Wade, R., Detzel, J.M., Adams, A.G., Buch, J.M., Beahm, D.L., Flask, C.A., Xu, K., et al. (2019). Functionalized Phenylbenzamides Inhibit Aquaporin-4 Reducing Cerebral Edema and Improving Outcome in Two Models of CNS Injury. Neuroscience 404, 484-498.

Giannetto, M.J., Gomolka, R.S., Gahn-Martinez, D., Newbold, E.J., Bork, P.A.R., Chang, E., Gresser, M., Thompson, T., Mori, Y., and Nedergaard, M. (2024). Glymphatic fluid transport is suppressed by the aquaporin-4 inhibitor AER-271. Glia.

Gomolka, R.S., Hablitz, L.M., Mestre, H., Giannetto, M., Du, T., Hauglund, N.L., Xie, L., Peng, W., Martinez, P.M., Nedergaard, M., et al. (2023). Loss of aquaporin-4 results in glymphatic system dysfunction via brain-wide interstitial fluid stagnation. eLife 12.

Huber, V.J., Tsujita, M., and Nakada, T. (2009). Identification of aquaporin 4 inhibitors using in vitro and in silico methods. Bioorg Med Chem 17, 411-417.

Igarashi, H., Huber, V.J., Tsujita, M., and Nakada, T. (2011). Pretreatment with a novel aquaporin 4 inhibitor, TGN-020, significantly reduces ischemic cerebral edema. Neurol Sci 32, 113-116.

Igarashi, H., Tsujita, M., Suzuki, Y., Kwee, I.L., and Nakada, T. (2013). Inhibition of aquaporin-4 significantly increases regional cerebral blood flow. Neuroreport 24, 324-328.

Jiang, R., Diaz-Castro, B., Looger, L.L., and Khakh, B.S. (2016). Dysfunctional Calcium and Glutamate Signaling in Striatal Astrocytes from Huntington's Disease Model Mice. J Neurosci 36, 3453-3470.

Mayo, F., Gonzalez-Vinceiro, L., Hiraldo-Gonzalez, L., Calle-Castillejo, C., Morales-Alvarez, S., Ramirez-Lorca, R., and Echevarria, M. (2023). Aquaporin-4 Expression Switches from White to Gray Matter Regions during Postnatal Development of the Central Nervous System. Int J Mol Sci 24.

Mola, M.G., Sparaneo, A., Gargano, C.D., Spray, D.C., Svelto, M., Frigeri, A., Scemes, E., and Nicchia, G.P. (2016). The speed of swelling kinetics modulates cell volume regulation and calcium signaling in astrocytes: A different point of view on the role of aquaporins. Glia 64, 139-154.

Risher, W.C., Andrew, R.D., and Kirov, S.A. (2009). Real-time passive volume responses of astrocytes to acute osmotic and ischemic stress in cortical slices and in vivo revealed by two-photon microscopy. Glia 57, 207-221.

Salman, M.M., Kitchen, P., Yool, A.J., and Bill, R.M. (2022). Recent breakthroughs and future directions in drugging aquaporins. Trends Pharmacol Sci 43, 30-42.

Shigetomi, E., Bushong, E.A., Haustein, M.D., Tong, X., Jackson-Weaver, O., Kracun, S., Xu, J., Sofroniew, M.V., Ellisman, M.H., and Khakh, B.S. (2013). Imaging calcium microdomains within entire astrocyte territories and endfeet with GCaMPs expressed using adeno-associated viruses. J Gen Physiol 141, 633-647.

Solenov, E., Watanabe, H., Manley, G.T., and Verkman, A.S. (2004). Sevenfold-reduced osmotic water permeability in primary astrocyte cultures from AQP-4-deficient mice, measured by a fluorescence quenching method. Am J Physiol Cell Physiol 286, C426-432.

Verkman, A.S., Binder, D.K., Bloch, O., Auguste, K., and Papadopoulos, M.C. (2006). Three distinct roles of aquaporin-4 in brain function revealed by knockout mice. Biochim Biophys Acta 1758, 10851093.